

# Formational Conditions of Ribbed Moraine in Norway: A Distribution Analysis and Ribbed Moraine Inventory

Thomas J. Barnes[1], Thomas V. Schuler[1], Karianne S. Lilleøren[1], Louise S. Schmidt[1]

[1]Department of Geosciences, University of Oslo, Oslo, 0371, Norway

*Correspondence to*: Thomas J., Barnes (thomas.barnes@geo.uio.no)

**Abstract.** Ribbed moraines are common landforms in regions formerly glaciated by the Fennoscandian, British and Laurentide ice sheets. Their process of formation is disputed, as formation conditions are hard to reconstruct. With this work, we address this issue through a combination of a comprehensive ribbed moraine inventory of mainland Norway, mapped topographic information, and modelled glacial and hydrological information. First, a detailed 10 metre resolution ribbed moraine dataset
is produced for the entire mainland Norway, which is used in combination with spatial statistics to isolate common distributions for ribbed moraines in nine series of data. These values include (a) topographic (elevation, slope and curvature), (b) glacial (basal temperature, ice thickness and ice velocity), and (c) hydrological values (flow accumulation, hydraulic head and hydraulic gradient). We derive mean conditions at 21000 years before present and compare values for areas of ribbed moraines and an equal area of land where ribbed moraines are not present. Our findings show that (a) ribbed moraines typically form in
flat, low gradient and low curvature depressions with a low hydraulic gradient, (b) hydraulic head, hydraulic gradient and ice velocity are globally important for ribbed moraine formation, while factors such as elevation and ice thickness are too spatially variable for a wide-scale link to be drawn, but they show a strong local relationship, (c) ribbed moraines are present in areas where ice flow was relatively slow, (d) occurrence of ribbed moraines in areas of high hydraulic head, low hydraulic gradient and low ice velocity suggests that ribbed moraines formed in transitional areas between slow and fast ice flow, which may
resemble a "patchwork" of slippery and sticky spots of high and low frictional resistance. However, these relationships are not definite, as we simply note relationships rather than process observations, and as such we conclude the possibility of an "equifinality" theory explanation for the formation of ribbed moraines.



## 1. Introduction

Throughout several regions formerly glaciated during the last glacial maximum, we find landforms known as "ribbed" or
"Rogen" moraine, which take the form of transverse ridges formed subglacially. Such features are documented in all previously
glaciated areas, such as the British and Laurentide ice sheets, and have been the subject of study since the mid-1900s (Hoppe,
1952; Frödin, 1954; Cowan, 1968; Lundqvist, 1969; Dunlop & Clark, 2006). Within Fennoscandia, ribbed moraines have been
documented along the former ice divide regions of the Fennoscandian Ice Sheet (FIS), most notably along the Norway-Sweden
border, and around the site of lake Rogen in Sweden, hence the common name of "Rogen moraine" (Lundqvist, 1969). Yet,
despite long-standing interest in these landforms, it is only since the 1990s that large-scale inventories of ribbed moraines have
been produced (Sollid & Sørbel, 1994; Dunlop & Clark, 2006; Trommelen, Ross & Ismail, 2014; Barnes et al., 2024), with
many of these not capturing individual landform morphometry. Ribbed moraines have been documented to take up a wide
range of spatial scales and morphological characteristics, summarised by Dunlop and Clark (2006) into 16 categories. From
this series of categories, we find ribbed moraines to generally have a series of morphological commonality: (a) they form
transverse to former ice flow directions, (b) they form in fields, often in close proximity to streamlined features, (c) they are
amplitude undulations within the landscape, and (d) they are frequently at the sub-kilometre scale (Dunlop & Clark, 2006;
Barnes et al., 2024). However, despite our reasonable understanding of ribbed moraine characteristics, their total extent and
process of formation are both less understood.

The formational processes of ribbed moraines have been a contested subject since their initial observation, with two core
schools of thought: first that they are monogenetic landforms which should have a singular process of formation (Trommelen,
Ross & Ismail, 2014; Dunlop & Clark, 2006), and second that they are a polygenetic landform type, with many potential paths
to formation (Möller & Dowling, 2018; Möller, 2016; Möller & Dowling, 2015; Finlayson & Bradwell, 2008), also known as
"equifinality". Within the former school of thought several hypotheses arise including the fracturing of till sheets (Sarala, 2006;
Hättestrand & Kleman, 1999; Hättestrand, 1997), modification of pre-existing features (Finlayson et al., 2010; Möller, 2006),
instabilities in the subglacial system (Fowler & Chapwanya, 2014; Dunlop, Clark & Hindmarsh, 2008; Aario, 1977) and shear
stacking of the basal till sheet (Lindén, Möller & Adrielsson, 2008). On the other hand, the equifinality studies tend towards a
sedimentological explanation, following aspects of the shear stacking theory and a process of melt-out of debris-rich basal ice
(Möller, 2006).

Despite the disagreement within the study of ribbed moraines, typically there are a series of agreed-upon commonalities in the
formational conditions of ribbed moraines, with these being that (a) ribbed moraines have some relation to warm-based ice,
(b) ribbed moraines form in relatively slow moving regions of an ice sheet (Trommelen, Ross & Ismail, 2014; Putniņš &
Henriksen, 2017) and (c) ribbed moraines are often present in proximity to streamlined landforms (Ely et al., 2016). In addition,
it is often suggested that basal meltwater is present at sites of ribbed moraine formation (Putniņš & Henriksen, 2017). Hence,
these theories identify a series of commonalities to investigate when considering the formational conditions of ribbed moraines.
As such, when combining our understanding of ribbed moraines from their prior detection, and formational theories, we



consider them to be an important landform for studying subglacial conditions of former ice masses. Their relationship to linear landforms, hydrology, and their potentially polygenetic origin suggests that they may be a central component of the subglacial system – hence in mapping and understanding the conditions under which ribbed moraines are present has the potential to widen our ability to visualise the processes occurring at the bed of former, and contemporary ice sheets.

Although the formation process of ribbed moraines is still not fully understood, several modelling studies attempt to address bedform ribbing under ice sheets. Dunlop, Clark and Hindmarsh (2008) attempted to constrain ribbed moraine characteristics using a theory of formation focusing on till sheet instability. Within this study, they note that ribbed moraines generally form at a short timescale, in the order of hundreds of years (Dunlop, Clark & Hindmarsh, 2008). Following this, both Ely et al. (2023) and Barchyn et al. (2016) attempt to constrain formational conditions using a numerical flow model. The former focuses

on temporal variation of basal landforms, whilst the latter focuses on variation based on ice flow velocity. In combination, both studies first add weight to the claim that ribbed moraines form in a relatively short time span and that they form in relatively slow ice flow velocities (200 m a$^{-1}$) (Dunlop, Clark & Hindmarsh, 2008; Barchyn et al., 2016; Ely et al., 2023). However, Barchyn et al. (2016) also identified that with a greater till thickness, ribbed moraines tended to persist for periods upwards of 200 years in their simulation, hence limiting the value of the temporal restrictions. In addition, Vérité et al. (2021)

used a physical experimental model setting and found that ribbed moraines form and persist in conditions with well-developed channelised water flow in the subglacial environment. This study additionally found that in situations where drainage was less channelised, ribbed moraines would be less persistent and often disappear within the short timescales other studies have found characteristic of the features. Thus, these prior studies show that hydrological conditions and ice velocity are key factors for constraining formational conditions using real-world examples.

Through this study we aim to use automatically mapped ribbed moraine data (Barnes et al., 2024) in conjunction with pre-existing mapped ribbed moraines (Kartverket, 2021) to produce a comprehensive Norway-scale ribbed moraine inventory. On this basis, we further perform spatial statistical analyses on formational conditions of ribbed moraines using contemporary terrain data (Kartverket, 2013), and modelled conditions during glaciation derived from a numerical reconstruction of the FIS (Fennoscandian Ice Sheet; Patton et al., 2016; 2017). Hence, we investigate the likely formational conditions of ribbed

moraines. We address the following objectives: (a) Identify and define all ribbed moraine fields in Norway, (b) Extract formational conditions for these fields to identify commonalities in formation, and (c) Use the results of (a) and (b) to narrow down and determine the most likely theories of formation for ribbed moraines, given the prevailing conditions. Hence, in performing this study we contribute to the debate on ribbed moraine formation and thus, provide insight into their formation based on a combination of modelled data and geomorphological data.

**1   Data & Methodology**

To carry out a study of ribbed moraine formational conditions, we first construct a comprehensive inventory of ribbed moraines in Norway. This country-scale inventory of ribbed moraines is produced both in vector format and raster format at 250 m



resolution. For this study, the inventory comprises of a series of ribbed moraine fields, rather than individual landforms due to the differences in scale magnitude. Specifically, our study takes data from Patton et al.'s (2016; 2017) ice sheet model, which

operates at a 10 km scale, which we interpolate down to 250 m. With data at an original scale of 10 km, it is unfeasible to draw comparisons to landforms at the scale of tens of metres, hence we consider ribbed moraine fields as distinct geomorphological sets. We secondly produce Norway scale datasets of formational condition data at 250 m resolution for the following groups of data: "Glacial conditions", "Morphological conditions" and "Hydrological conditions".

## 1.1 Ribbed Moraine Inventory

We produce the ribbed moraine inventory from two primary sources. We initially employ the methodology developed in Barnes et al. (2024) and iterate on a national scale at 10 m resolution. Hence, we apply this method to $3.24 \times 10^5$ km$^2$ of clustered data, and obtain a clustered dataset. Following this, we carry out manual quality control on the output data through comparison to the Norwegian national DEM (Kartverket, 2013) and high-resolution aerial imagery (Kartverket, 2023). This ensures that we work with high-quality and accurate landform polygons for Norway. Due to the limitations of the methodology in Barnes

et al. (2024), where detection accuracy reached a maximum of 70-75%, we merge this data with nationally mapped landforms (Kartverket, 2021). In doing this, we address the concerns of missing ribbed moraines from the dataset, and produce a comprehensive inventory of ribbed moraines in Norway. This merging was done by collecting data from the Norwegian national superficial geology data (Kartverket, 2021) and extracting all ribbed moraine polygons. Then, using a polygon merge tool, we combined each ribbed moraine dataset, hence resulting in a comprehensive national dataset of ribbed moraine

landforms in Norway.

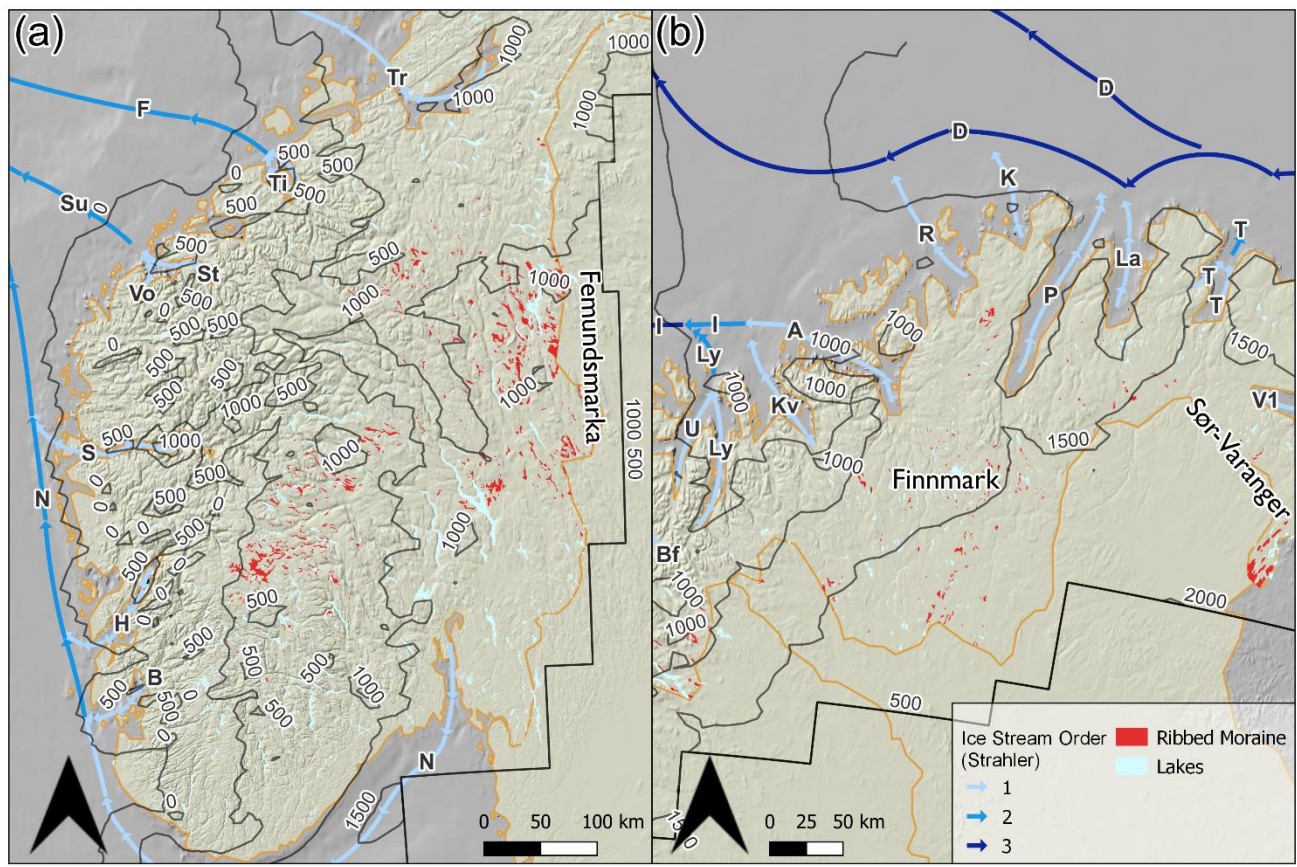

**Figure 1: Overview map of ribbed moraine fields (red) inventory in (a) southern and (b) northern Norway. Background shaded relief based on data from the Norwegian mapping authority (Kartverket, 2016; Kartverket, 2007). Lakes larger than 1 km² (NVE, 2019) are shown in (light blue). Contours (black) indicating ice sheet thickness at 21 ka BP according to Patton et al. (2016; 2017).**
**Ice stream information from Montelli et al., (2017) and Ottesen et al., (2008), streams are ordered by colour using a Strahler stream order. Ice stream labels indicate their name as follows: V = Varangerfjorden, T = Tanafjorden, La = Laksefjorden, P = Porsangerfjorden, R = Revsbotn, A = Altafjorden, Kv = Kvænangen, Ly = Lyngen, I = Ingøydjupet, D = Djuprenna, K = Kobbefjorden, N = Norwegian Channel, Su = Sulafjorden, S = Sognefjorden, F = Frøyhavet, Tr = Trondheimsfjorden, U = Ullsfjorden.**

However, due to the scale differences between ribbed moraines (order of 10 metres) and the scale of glacial models such as Patton et al. (2016; 2017) (order of 10 kilometres) and the computing costs of iteration across all of Norway, we settle on two compromises in data quality. We determine that it is more valuable to sample ribbed moraine fields, hence considering the spatial area between ribbed moraine landforms as an important part of their morphology (Dunlop & Clark, 2006; Sarala, 2006); thus, we produce a ribbed moraine "field" inventory. This is done through calculation of minimum bounding polygons around

fields of ribbed moraine, thus defining each group of landforms with a single polygon. Our ribbed moraine field inventory results in a series of 762 mapped ribbed moraine fields in mainland Norway. Our second compromise addresses the resolution disparity more directly, as we resample all data, including rasterised ribbed moraine fields to 250 m resolution. This allows for




relatively rapid data processing, whilst maintaining a higher resolution than modelled from Patton et al. (2016; 2017). As a result, we produce a Norway-scale ribbed moraine dataset at 250 m, comprising of 3950 (x) by 6402 (y) pixels. However, due

to the Patton model originating at 10 km resolution, it is impossible to derive ice conditions at a scale below 10 km, rather we downscale 10 km values to 250 m for easy of georeferencing. As a result of this, it is important to consider that sub-10 km changes ice conditions may not be well resolved in the data, for example changes in ice thickness from summit to valley basin.

## 1.2 Formational condition data

Formational conditions are gathered from ice sheet model data for the Fennoscandian Ice Sheet (Patton et al., 2016; 2017),

and morphological data is collected and derived from Norway's national 50 m elevation model (Kartverket, 2007). Further processing of data occurs within QGIS and "TopoToolbox" (Schwanghart & Scherler, 2014) in MATLAB. In total, we calculate eight formational condition datasets for the analysis of ribbed moraines: three glaciological values (basal temperature, ice flow velocity, and ice thickness), three physical values (elevation, slope and curvature) and three hydraulic values (hydraulic head, hydraulic gradient, and subglacial flow accumulation).

### 1.2.1 Glaciological data


Using the glaciation model described in Patton et al. (2016, 2017), we derive a series of glaciological conditions for every 1 ka BP timestep between 11 ka BP and 25 ka BP. These conditions include: basal temperature (°C), ice flow velocity (m a$^{-1}$) and ice thickness (m). These data are sampled for the full spatial extent of Norway's 50 m elevation data, by clipping the model data to the limits of a merged raster of all 50 m elevation data, and resampled from 10 km to 0.25 km resolution using GDAL

warp's cubic resampling algorithm. We use the same process as a means of resampling up from 50 m to 250 m with the elevation data and its derivatives below in section 2.2.2. These variables are selected in reference to prior literature on ribbed moraine formation, specifically concerning their formation in low ice flow velocities (Ely et al., 2023; Barchyn et al., 2016), and in the presence of basal water (Trommelen, Ross & Ismail, 2014) or temperate conditions (Trommelen, Ross & Ismail, 2014; Möller, 2006). Furthermore, the use of ice thickness and basal temperature allow for a calculation of pressure melting,

and thus, basal thermal conditions at each timestep (S. 2.3.2).

### 1.2.2 DEM derived data

We gather the 50 m elevation model from Norway's national data repository (Kartverket, 2007). These data are resampled to 250 m resolution for all of mainland Norway. Furthermore, we derive two morphometric characteristics from the DEM dataset: slope at 250 m resolution, and secondly total curvature at 250 m resolution. We derive these values using the topotoolbox

MATLAB library (Schwanghart & Scherler, 2014). We use slope and curvature as these were key masking and morphometric DEM derivatives used in Barnes et al., (2024) for ribbed moraine detection. We include DEM in the analysis to determine elevation ranges for each moraine field, due to former literature stating elevational limits for ribbed moraines in Fennoscandia (Sommerkorn, 2020; Sollid & Sørbel, 1994).





### 1.2.3 Hydraulic characteristics

From the DEM data and the glaciological data, we calculate three subglacial hydraulic characteristics: subglacial hydraulic head (m of water), subglacial hydraulic gradient (m m⁻¹) and subglacial flow accumulation (m² in upstream area) using the MATLAB-based TopoToolbox (Schwanghart & Scherler, 2014). We calculate subglacial hydrological values through the standardised equations (Eq. 1; Eq. 2) built-in to TopoToolbox tools.

**Equation 1:**

$$h_{head} = h_p + h_z$$

**Equation 2:**

$$h_p = \frac{k * \rho_i * H}{\rho_w}$$

**Equation 3:**

$$h_{grad} = \frac{(h_1 - h_2)}{L}$$


Where $h_p$ is the pressure head (m) and $h_z$ is the elevation head (m) for equation 1, and where $h_1$ & $h_2$ are points along distance $L$, for equation 3. However, to simulate subglacial conditions, we include the subglacial pressure derived from ice overburden pressure; and we simulate realistic subglacial topography by calculating isostatic depression at each timestep of the Patton model (Patton et al., 2016; 2017). We base this calculation on the assumption that water pressure is a constant fraction of the

ice overburden pressure, as in equation 2, where $H$ is ice thickness (m), $\rho_i$ is ice density (917 kg m⁻³), $\rho_w$ is water density (1000 kg m⁻³) and $k$ (0.9) is the flotation fraction.

### 1.2.4 Basal Temperature Analysis

Using the basal temperature values for 21 ka BP from Patton et al. (2016; 2017), we assess whether the basal temperature is at or below the pressure melting point. The melting temperature for a given ice overburden pressure is given by equation 4:

**Equation 4:**

$$T_{mp} = \gamma * \rho_i * g * H_i$$

Where $T_{mp}$ is pressure melting point in °C, $\rho_i$ is ice density (917 kg m⁻³), g is gravitational acceleration on earth (9.81 m s⁻²), $H_i$ is ice thickness (m), and $\gamma$ the melting temperature depression coefficient (°C Pa⁻¹), which ranges between 7.4x10⁻⁸ and

9.8x10⁻⁸ °C Pa⁻¹, for pure ice and air free water, versus pure ice and air saturated water respectively (Aschwanden, 2010). For example, if there was 1 km of ice present, then the basal melting point can range between -0.67 - -0.88 °C using the aforementioned values.



### 1.3 Spatial Analysis

To analyse formational conditions, we followed four key steps: (a) division of Norway into study areas (S. 2.3.1), (b) identifying the most likely period of formation of ribbed moraine fields (S. 2.3.3), (c) balanced sampling of data and (d) analysis of value distributions between ribbed moraine fields and glacial, hydrological and geomorphological information, versus the distribution of values for areas outside moraine fields. Each step was constructed and iterated in MATLAB using data from S 2.1 & S 2.2.

#### 1.3.1    Study Area Subdivisions

As glaciological conditions differed throughout Norway during the presence of the FIS, we subdivide the country-wide analysis into logical subdivisions. As such, we discriminate North Norway, South Norway, and further subdivided South Norway into "under 900 m" and "over 900 m". The North-South subdivision was selected as 66 °N, which split our ribbed moraine inventory without dissecting any fields (Fig 2). A north-south divide in Norway is already present in regard to topography (Etzelmüller, Romstad & Fjellanger, 2007), ribbed moraine (Barnes et al., 2024) and ice conditions (Patton et al., 2016; 2017), and hence was the first most logical step. Secondly, when analysing elevational distribution of ribbed moraines in South-Norway, we identified the mean value as between 800-900 m elevation. This elevation marks a distinct division in the data (Fig. 2) - while also clearly dividing our ribbed moraine inventory into two main sets. These sets are the inland set (over 900 m elevation) which are present on the edge of the central Norwegian mountains (Fig. 2), and the border set, which lay along the Swedish border within Femundsmarka (Fig. 2), an area in proximity to the Swedish "Lake Rogen" region (Lundqvist, 1969).





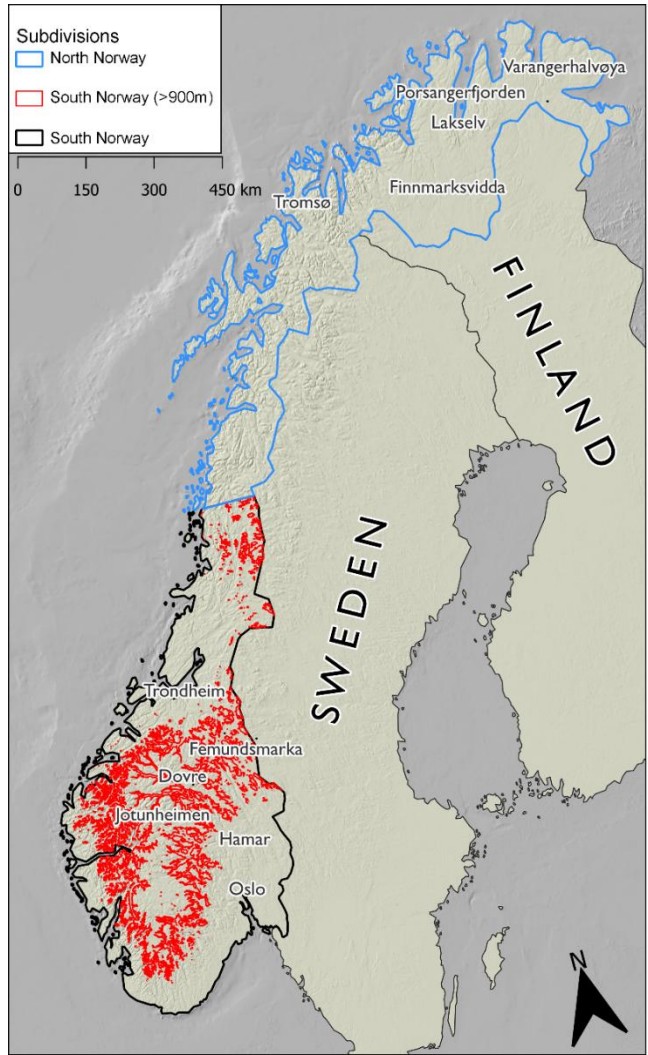

**Figure 2: Our subdivisions for this study, derived using the Norwegian national height model (Kartverket, 2007). No elevation distinction is made in the north of Norway versus the south due to the scarcity of ribbed moraine and the lack of variability within our results for north Norway.**

### 1.3.2    Data Balancing & Sampling

Due to the disparity in the area of ribbed moraines versus non-ribbed moraine areas, we follow a data balancing method to produce a dataset of even samples for all data layers. For each study set, we count the pixels classified as ribbed moraine. This value defines the number of values to be sampled from the entire "non-ribbed moraine" dataset at similar elevation (mean of $z_{rm} \pm 1.5 \, \sigma$) to produce a set of evenly balanced samples. Furthermore, we sample for the period of 21 ka BP as it is during this timestep when ribbed moraine line up most consistently with modelled ice flow direction (Patton et al., 2016). In addition, we sample for the period of 14 ka BP as modelled ice flow also matches here, however, due to the boundaries of the ice sheet it



makes sense that many ribbed moraine are not formed during this period. Hence, we produce a dataset over each of the two epochs – each including all mapped ribbed moraines, all areas without moraines following the same spatial distribution and
with the same number of samples. In doing this, we produce a paired dataset of areas with and without ribbed moraines of equal area.

### 1.3.3 Temporal comparison of the 21 ka BP flowset to 14 ka BP

In testing our data we carried out a comparison between 21 ka BP and 14 ka BP, both periods in which modelled ice flow direction match most closely with being "perpendicular" to ribbed moraines mapped across Norway. In this comparison, we
find more problems with the 14 ka BP period, as many ribbed moraine are present beyond the modelled margins of the ice sheet, and in many cases, the ice is far too thin for subglacial water to be present due to a lack of basal pressure melting. Despite this, we note the values for 21 ka BP and 14 ka BP show similar patterns, possibly due to model scale, and hence determine that it is more reasonable to use the 21 ka BP period for study, due to the full coverage of the ice sheet. Furthermore, as 21 ka BP is still during the period of deglaciation (Patton et al., 2017; Stroeven et al., 2016), whilst maintaining thick enough
ice for pressure melting, as suggested in the subglacial drainage from Femundsmarka (Fig. 2) down into the Atlantic via Europe described in Patton et al. (2017), we provide our analysis using this period. This is aided by Putniņš and Henriksen (2017), who suggest ribbed moraine to form during the slow down and speed up of an ice sheet, often during a transition from large-scale to locally-dominated flow regimes, or in areas of patchy warm and cold based ice. The "patchy" warm and cold based ice theory further connects with Shackleton et al.'s (2018) theory where subglacial thermal regime is strongly controlled by
local variations in ice conditions.

### 2 Results

We analyse the total population of ribbed moraines in Norway with regard to the above defined set of characteristics and then extend this analysis to different subsets of this population. Whilst there is a difference between areas with and without ribbed moraine for the elevation data, this is more likely due to the mean elevations of the sites where ribbed moraines have enough
space to form in Norway than any formational process. Hence, our results can be split into topographical (slope and curvature), hydraulic (hydraulic head and hydraulic gradient) and glaciological data (basal temperature, ice thickness and ice flow velocity). Our equal area samples of ribbed moraines as defined in S.2.3.3 and areas without ribbed moraines were tested with several model iterations selecting a random population of areas without ribbed moraines. This means of testing our data showed that regardless of iteration, the results had little variance, and thus we are confident in the validity of our samples. We display
our samples comparatively as violin charts (Figure 3, 4, 5) (Bechtold, 2016), as they provide a clear visualisation of data distribution without assuming a specific statistical distribution. From these, we can understand if there are normal or multimodal distributions within our data, and thus if there are distinct populations of ribbed moraine throughout our dataset.



## 2.1 Ribbed moraine inventory overview

Our count of ribbed moraine pixels results as 49524 pixels in Norway as a whole, divided into subgroups of 8061 pixels in

north Norway and 41492 in south Norway. Pixels in south Norway above and below 900 m were roughly equally divided. As a result of this, we can confirm that there are enough pixels of ribbed moraine in each region to carry out a detailed and statistically significant analysis, however it is important to note that our results from southern Norway will be more easily confirmed, whilst in northern Norway due to the fewer pixels, our results may be less concrete. This distribution of pixels further indicates that ribbed moraines, whilst present in northern Norway, are not as common in comparison to southern

Norway. This may simply be a factor of national borders, as many ribbed moraines and ribbed moraine adjacent features such as *murtoos* are common across the border into Finnish Lapland (Ahokangas et al., 2021).

## 2.2 Topographical information

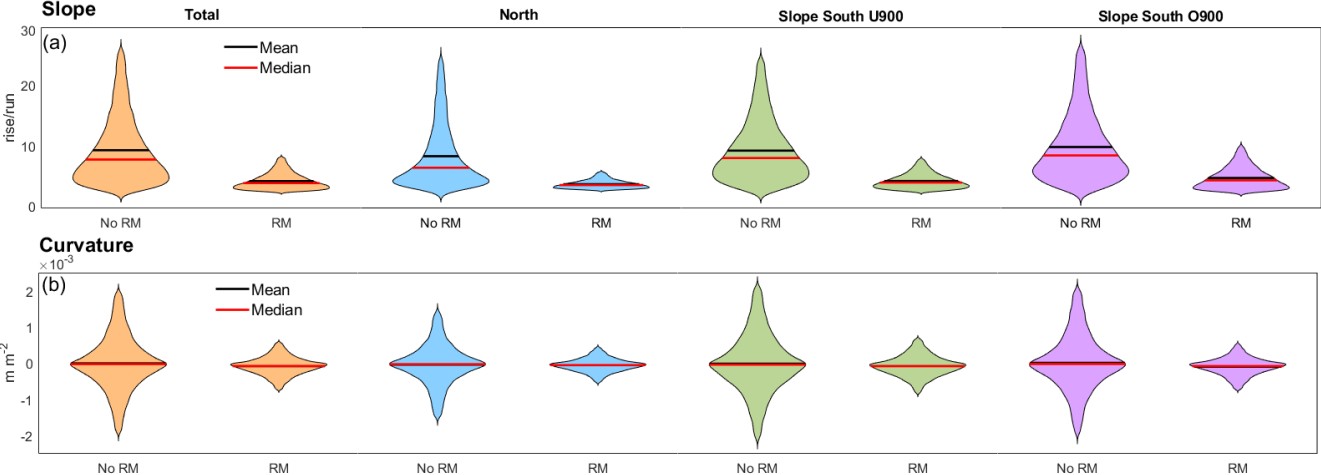

**Figure 3. Violin distribution plots of (a) slope and (b) curvature for areas of no ribbed moraine presence (No RM) and ribbed moraine presence (RM). Colours represent spatial distributions, with orange, blue, green and purple being all of Norway, north Norway, south Norway under 900 m elevation and south Norway over 900 m elevation respectively. Red and black lines represent median and mean values. Number of samples for each region are (a) total: 49524 points, (b) north: 8061 points, (c) south u900: 26283 points, (d) south o900: 15209 points.**

The distributions of both slope and curvature are consistent between areas with ribbed moraines and without ribbed moraines, however to different extents. It is, however, clear that distributions within the ribbed moraine features are narrower than those of areas without ribbed moraines – which thus indicates a greater degree of homogeneity in the ribbed moraine data versus the non-ribbed moraine data. Both sets of distributions show unimodal distributions, however slope demonstrates a skewed distribution towards zero, curvature is normally distributed above and below zero (Fig. 3a, b). Slope's skewed distribution (Fig.

3a) demonstrates that the majority of Norway has areas of low slope, however, ribbed moraines are concentrated only in these



areas, with a maximum rise/run value of $8.9 \times 10^{-2}$ m m$^{-1}$, versus a maximum value in areas without ribbed moraines of nearing 30 m m$^{-1}$. This varies somewhat between each regional distribution, with the peaks in the distributions for north Norway being sharper than in other regions, and the lowest maximum value of $4.2 \times 10^{-2}$ m m$^{-1}$. Adding to this, we note that whilst the distributions of slope in ribbed moraine areas is distinctly much narrower than areas without ribbed moraines, the distribution

still typically lies close to the majority of values in areas without ribbed moraines. This is notable as in all distributions, the peak value for areas without ribbed moraines corresponds to the mean value for areas with ribbed moraines. Hence, we note ribbed moraines to be concentrated in areas of low slope, but not outside the majority of values for Norway in general.

This pattern is reflected in the values for curvature, where the maximum and minimum values of curvature in areas of ribbed moraines range from $-0.9 \times 10^{-3}$ m m$^{-2}$ to $0.9 \times 10^{-3}$ m m$^{-2}$, whereas areas without ribbed moraines reach more than double these

values ranging between $-2.2 \times 10^{-3}$ m m$^{-2}$ to $2.2 \times 10^{-3}$ m m$^{-2}$ (Fig. 3b). In this comparison we note an almost identical situation to that demonstrated by slope, but to a greater degree. Areas where ribbed moraines are present are concentrated in the low end of curvature values for the general landscape of Norway, however these areas are also within the core of the distribution of curvature values for the general landscape of Norway.

Spatial variations between regions in this case are relatively minor, with only slight differences from region to region.

Distributions appear to be more or less concentrated around the peaks, with the majority of Norway, and the areas under 900 m in southern Norway showing the greatest distinction between peak distribution and the extremities. This can be seen in the greater concavity between maximum and minimum distributions (Fig. 3). Spatial variability in distribution is also apparent, particularly between the north of Norway in slope, as the distribution is highly concentrated, with a range of only $3.7 \times 10^{-2}$ m m$^{-1}$, in contrast to the mean range of $7.0 \times 10^{-2}$ m m$^{-1}$ and the maximum range in the south below 900 m elevation of $9.9 \times 10^{-2}$ m

m$^{-1}$. Similar patterns are harder to identify in curvature, with only the north of Norway showing clear distinction from the other groups, but still lying less than 1 standard deviation from the mean range.

## 2.3 Hydraulic information

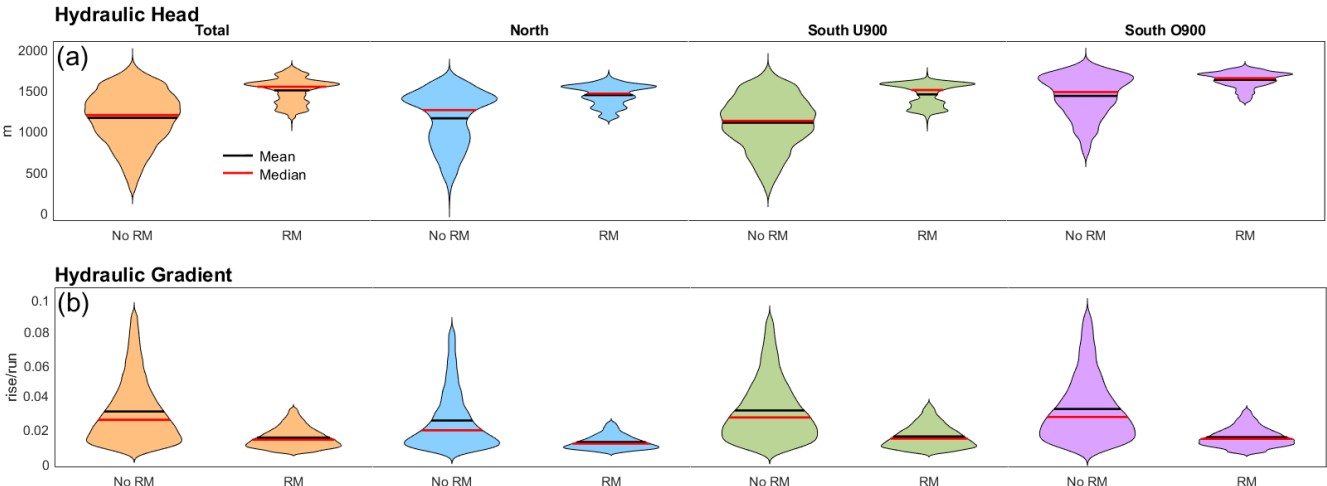



**Figure 4. Violin distribution plots of (a) hydraulic head and (b) hydraulic gradient for areas of no ribbed moraine presence (No RM) and ribbed moraine presence (RM). Colours represent spatial distributions, with orange, blue, green and purple being all of Norway, north Norway, south Norway under 900 m elevation and south Norway over 900 m elevation respectively. Red and black lines represent median and mean values. Number of samples for each region are (a) total: 49524 points, (b) north: 8061 points, (c) south u900: 26283 points, (d) south o900: 15209 points.**

### 2.3.1 Hydraulic head

Hydraulic head distributions depart from the commonalities between ribbed moraines and non-ribbed moraine areas seen in the physical sets of data (Figure 4). We note in all cases only 10% of ribbed moraine values are present below the mean of the non-ribbed moraine areas, like with the topographical values, yet in addition the patterns in distribution vary drastically. All non-ribbed moraine areas excepting north Norway demonstrate a unimodal distribution, whilst we see a bimodal distribution in the North that is not reflected in the ribbed moraine values (Fig. 4a). This is likely due to the two predominant types of landscape in north Norway – high elevation mountainous terrain, and moderate to low elevation coastal tundra. Our analysis shows moraines exist in areas of high hydraulic head (h > 1000 m) across Norway as a whole, with a mean peak in distribution at 1569 m of hydraulic head and all peaks being within ±100 m of the mean. Additionally, when comparing the mean of ribbed moraines and non-ribbed moraines, we note that ribbed moraines are present where hydraulic head is 292 m greater than non-ribbed moraine areas, an increase of 24%. Hence, it can be clearly defined that ribbed moraines exist in areas with a high hydraulic head relative to the general format of the Norwegian landscape. Yet, like with the physical values, ribbed moraines are typically present around the greatest concentration of points in the general landscape, meaning that they form within areas that are not determined by hydraulic head alone.

Within this analysis, we note that the values in the south of Norway under 900 m have less distinct variation from the whole population versus values from the North and South over 900 m values. South over 900 m expresses an extreme skewing to the high hydraulic gradient values, with its mean lying ~100 m higher than the mean of dataset covering all of Norway. Furthermore, the spread of data is much more compressed than in other regions. On the other hand, we identify a pattern of a multimodal distribution in the north of Norway, with four peaks in the data at roughly equidistant intervals throughout its hydraulic head range. Despite the bimodal layout of the non-ribbed moraine values in north Norway, we find all four peaks of ribbed moraine values in the north lie above the trough between the upper and lower peaks in the data covering areas without ribbed moraines (Fig. 4).

### 2.3.2 Hydraulic gradient

The distributions of hydraulic gradient predominantly mirror the distributions in slope, but to a smaller degree. Whilst slope values range from 0 – 0.28 m m$^{-1}$, hydraulic gradient ranges from 0 – 0.09 m m$^{-1}$. Despite this disparity in scale, the patterns of distribution remain largely similar, with only minor differences in the south over 900 m data and the north Norway data, where the former has a wider distribution towards the low-end of gradient, and the latter has a wider relative distribution from minimum to maximum (Fig. 3a, 4b). As a result, there is evidence from the hydraulic data that ribbed moraines are restricted to areas of very low modelled hydraulic gradient ($h_{grad} < 0.04$ m/m), but high hydraulic head ($h_{head} > 1000$ m) when compared





to the distributions for areas without ribbed moraines in Norway. Furthermore, this is mirrored by the relationship between slope and ribbed moraine presence.

## 2.4 Glaciological information

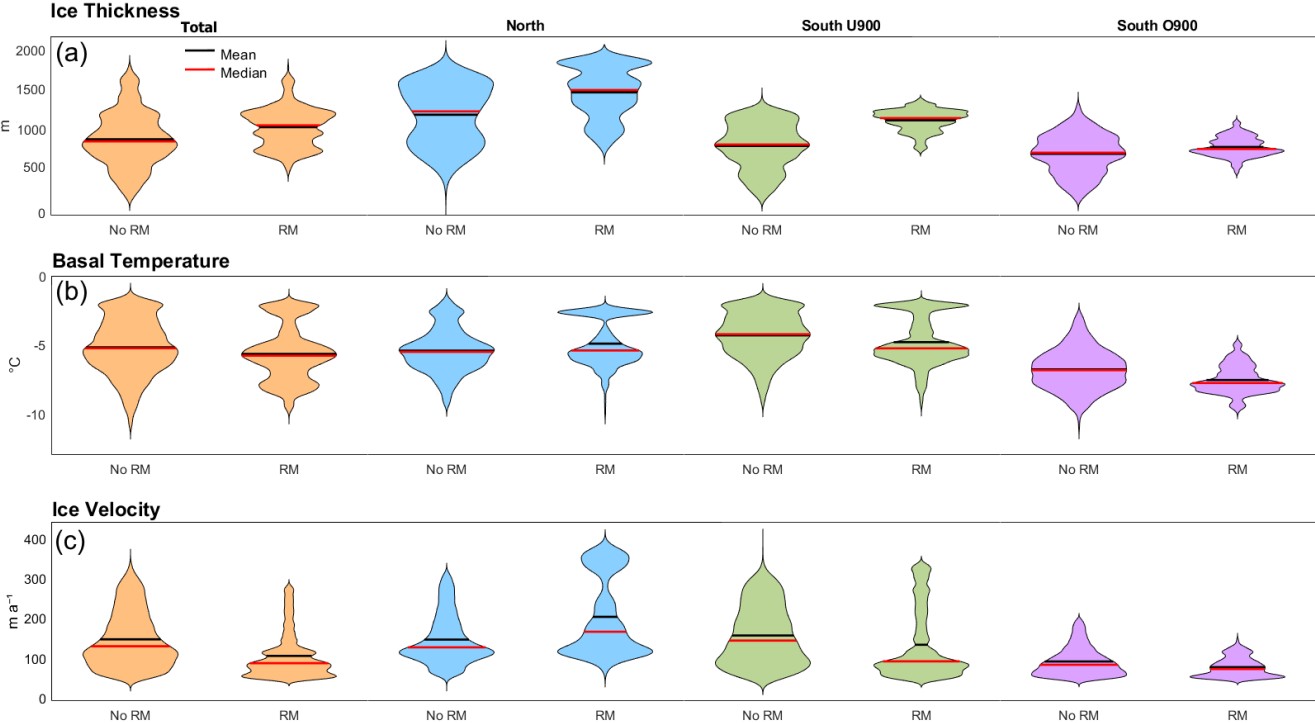

**Figure 5. Violin distribution plots of (a) ice thickness, (b) basal temperature and (c) ice velocity for areas of no ribbed moraine presence (No RM) and ribbed moraine presence (RM). Colours represent spatial distributions, with orange, blue, green and purple being all of Norway, north Norway, south Norway under 900 m elevation and south Norway over 900 m elevation respectively. Red**
**and black lines represent median and mean values. Number of samples for each region are (a) total: 49524 points, (b) north: 8061 points, (c) south u900: 26283 points, (d) south o900: 15209 points.**

### 2.4.1  Ice thickness and basal temperature

Ice thickness shows a large range in the ribbed moraine data (Figure 5a) which closely corresponds to spatial distribution. The ribbed moraine data has a trimodal distribution for the entirety of Norway, each peak referring to a specific spatial subsection
of Norway, where the upper peak of distribution at 1197 m refers to a combination of North Norway and South Norway under 900 m elevation, the middle peak refers to southern Norway under 900 m elevation at 926 m ice thickness and the lowest peak refers to southern Norway over 900 m elevation at 688 m ice thickness. This is made clear through comparing the ice thickness means of each area's ribbed moraines to the total violins (Fig. 5a). This shows that ice thickness is controlled by elevation, but also, that ice thickness can be important as a spatial predictor, rather than a global predictor given the wide range of distribution
bands in our output. In northern Norway, the ribbed moraines tend to be present at higher modelled ice thicknesses compared



to those in southern Norway, which further adds evidence to their value as regional predictors, rather than global predictors for ribbed moraine presence.

Within each region of Norway we further see several peaks, most notably in northern Norway, where a trimodal distribution is evident, with the most distinct peak at values approaching ~2000 m ice thickness. These values are found in the low-lying
Sør-Varanger region of northern Norway, whilst the remaining ribbed moraines are located within the middle-altitudes of Finnmarksvidda (Fig. 1).

Basal temperatures for the full-scale dataset exhibit a trimodal distribution with peaks at -1.8 °C, -4.8 °C, and –7.0 °C. These correspond to distributional peaks in north Norway, south Norway under 900 m elevation, and south Norway over 900 m elevation respectively, meaning high elevations in south Norway have the lowest modelled basal temperatures.

When considering each region in isolation, and using equation 4 to calculate the pressure melting point at the bed of the thickest modelled ice in each region, we note that the lowest possible pressure melting points range between -1.42 °C to -1.87 °C for North Norway, -0.78 °C to -1.03° °C for  South Norway over 900 m elevation, and -0.97 °C to -1.28 °C for South Norway under 900 m elevation. This shows that, according to Patton et al.'s (2016; 2017) model, all areas have colder average values than the lowest possible pressure melting threshold. However, whilst the average value from Patton et al., (2016; 2017) appears
accurate for ice thickness, local scale, sub-grid variation (<10 km) is unresolved, thus leaving room for elevation and, by association, ice thickness variation of over 1000 m within each grid cell. In calculating pressure melting point values for Norway and identifying where temperatures are modelled to be above these values, we find areas in proximity to ribbed moraine concentrations which are modelled as warm based ice. When taking note of the ice thickness variations over the spatial extent of each grid cell, it is not unfeasible that, as ribbed moraines are present in depressions in the landscape, they may be
present in local patches of warm based ice that are unresolved by the model.

### 2.4.2    Ice flow velocity

We find ice flow velocity to show value as a global predictor for ribbed moraines rather than a regional predictor as distributions of ice flow velocity in areas of ribbed moraines show a consistent pattern of low velocities (Figure 5c). Typically, we note ribbed moraines to form in areas of slower ice flow than areas without ribbed moraines. This is mirrored through prior
literature, with Ely et al., (2023) and Barchyn et al., (2021) both suggesting ribbed moraine formation occurring in slow ice flow regions based on numerical modelling approaches. We see this clearly in the mean values for all of Norway, where areas without ribbed moraines have a mean ice flow velocity of 120 m a$^{-1}$, whereas areas with ribbed moraines have a mean ice flow velocity of 90 m a$^{-1}$, mirroring values in literature (Ely et al., 2023; Barchyn et al., 2021). The violins further show a concentration of ribbed moraine pixels in the lower end of each distribution (Fig. 5c), with median values of 35 m a$^{-1}$ for south
Norway under 900 m elevation and 38 m a$^{-1}$ for south Norway over 900 m elevation. We see a wider distribution of ice velocities in the north and in the south under 900 m elevation, with an extended distribution up to 358 m a$^{-1}$ and 289 m a$^{-1}$ respectively and a more pronounced peak in distribution in the north. In the north, this corresponds somewhat to ribbed moraines in Sør-Varanger (Fig. 1), which are in local proximity to the Varangerfjorden ice stream (Fig. 1), which could indicate





ribbed moraines being present at the head of the ice stream, thus evidencing the high ice flow velocities in the upper peak of
the northern dataset. However, due to north Norway being a smaller component of our data (only 16% of total ribbed moraine
pixels), the higher values in north Norway can be seen as outliers or minor versus the total study. In southern Norway, we do
not see a similar concentration, but a wider extent of ice flow velocities in the areas under 900 m a$^{-1}$. These values correspond
to areas outside of Femundsmarka (Fig. 1), but also areas along the Norway-Sweden border where we clip the model, perhaps
leading to these discrepancies. Alternatively, these areas could be related to the Oslofjorden ice stream (Fig. 1). Despite some
variability however, we note that ice flow velocities across Norway in areas of ribbed moraines trend towards sub-120 m a$^{-1}$
flow velocities on average, whilst the overall ice flow velocities are much more distributed, between $0 - 300+$ m a$^{-1}$.

## 2.5  General overview of data distributions

When sampling values for all of Norway, we observe differences in the distributions of ribbed moraine and non-ribbed moraine
areas, as well as distinct peaks within each investigated value. These peaks often result from spatial differences, which aligns
to some extent with the three regions defined in figure 2 (North Norway, South Norway under 900 m elevation and South
Norway over 900 m elevation. However, while most values show multimodal distributions that are connected to spatial
distribution, some multimodal distributions are less distinct, for example ice velocity and hydraulic head. Moreover, slope,
curvature and hydraulic gradient demonstrate distinct unimodal distributions.

While ribbed-moraine areas are found in many different locations in Southern Norway, there are only two areas in North
Norway in which ribbed moraine are found: "Finnmarksvidda" (Fig. 1), a moderate elevation, rolling moorland landscape, and
Sør-Varanger (Fig. 1), a low elevation wetland and tundra landscape. Whilst the elevational distribution of ribbed moraine is
not directly related to their formational processes due to the limited landscape available for ribbed moraine formation in
Norway, we note that there are different elevational bands in each region of the country where ribbed moraines are present.
For northern Norway, there are clear populations of ribbed moraines at 125 m, 450 m and 625 m, with the former corresponding
to Sør-Varanger, and the latter corresponding to Finnmarksvidda.

## 3    Discussion

In this section, we first outline the common conditions of our mapped ribbed moraine inventory, and from this we infer the
more and less valuable information for general versus regional ribbed moraine detection. Next, we analyse the spatial
distribution of ribbed moraine, and through literature infer the presence of warm subglacial conditions during the FIS. Finally,
we address the prevailing theories of ribbed moraine formation, coupling our data with literature to determine common
formational conditions and evaluate the "most likely" theories of formation.



### 3.1.1 Addressing spatial variability in the ice model

Whilst our results show that at 21 ka BP areas with ribbed moraines generally have temperatures too cold for pressure melting, several points may be used to address this and thus allow for the presence of subglacial water. These points primarily arise
from inaccuracies in the glaciation model we use for this study (Patton et al., 2016; 2017). First and foremost, the primary contribution to inaccuracy in this model is the spatial resolution. We note that the glacier model we use for this study runs at a 10 km resolution, and thus detail smaller than 10 km is unresolved in our outputs. This is important, as ribbed moraine fields may often be longer than 10 km, but they typically are situated in depressions in the landscape (Dunlop & Clark, 2006), which are typically narrower than 10 km in width. These depressions, and by association, higher peaks than the average elevation of
each grid cell, can lead to elevation variations by > 1000 m over the area of a single grid cell. As such, it is not unlikely that areas which are close in temperature to the pressure melting point (on average), may have a patchwork of deeper valleys where basal temperatures are at the pressure melting point, and thus basal water can be present. In addition to the spatial scale-derived inaccuracy, we note that the model used does not address frictional melting and strain-based heating processes within the ice mass, and thus the basal temperature in areas before obstructions in flow are likely lower than the true values. These two
factors combine to present the possibility of there being water at the bed of the FIS, particularly in areas where ribbed moraines may have been common due to their presence in depressions at the glacier bed. Such effects from the variability within a 10 km cell may only just tip temperatures over the melting threshold, but this would be sufficient for a theory focusing on a patchwork of water-filled cavities. The presence of water in this capacity would link somewhat to Patton's map of potential subglacial drainage (Patton et al., 2017), which includes flow from the Femundsmarka region (Fig. 2) out via Europe and into
the Atlantic at 22.7 ka BP during the onset of deglaciation. As ice conditions within the Femundsmarka region did not change to a great degree between 22.7 and 21 ka BP (Patton et al., 2017) it is possible that temperate conditions persisted. Hence, whilst ice thickness and basal temperatures show conditions where the FIS maintains a cold-based setting, we suggest that the model is not high enough in resolution to rule out the possibility of cavities of temperate basal ice due to increased thickness within grid cells, and thus hydrological influences cannot be ruled out - particularly in areas with high elevational variation
such as Norway. As such, we continue the discussion with this point in consideration.

### 3.1.2 Distribution within our mapped values

Within our study of ribbed moraine formational conditions, we note two key categories:

1. values which are strong predictors within the entire dataset, and thus show the presence of ribbed moraine within a narrow band of values versus the total dataset;
2. regionally specific predictors, which are strong regionally but not globally.

The simple narrow-band global predictors are slope and hydraulic gradient, two similar values, and curvature. The two more complex narrow-band global predictors are hydraulic head and ice velocity, which both have ribbed moraines occupying a narrow band of the total data, outside of the core of the main dataset, and with some regional variation. The narrow band values





are spread across the types of data – two geomorphological variables, two hydrological variables and one ice dynamics
variable, suggesting that no single aspect of glacial characteristics is responsible for ribbed moraine formation. This theme is
common with the regionally specific values – elevation, ice thickness and basal temperature, which further adds to this,
meaning that ribbed moraine are a product of the complex interaction of several systems within the glacier. However, we
address regionally specific values as much less useful in this study, as our aim is to identify factors which can be used to predict
ribbed moraines in a general sense, or rather, factors that indicate the formational conditions of ribbed moraine.

Curvature and slope appear to be key in the formation of ribbed moraine as ribbed moraines tend to form in areas of both low
slope and curvature. Barnes et al. (2024) identifies this general characteristic as "general slope", whilst Sommerkorn (2020)
also notes the common presence of ribbed moraine on areas of sub-5 degree gradient slopes, and by association, low curvature.
These concepts are further reflected in literature with Dunlop and Clark (2006) identifying ribbed moraine as typically forming
in valleys, basins and small topographic depressions, while less commonly in convex regions such as hillslopes. Hence,
geomorphological evidence shows that in general the presence of relatively flat topographic depressions can be used to detect
ribbed moraine – and thus are a factor in their formation.

Coupled with this, we note the hydrological factors in ribbed moraine formation of hydraulic head and hydraulic gradient, the
latter being closely related to slope. Hydraulic head can be used as a parallel for the subglacial water pressure, based on ice
thickness and ice bed elevation, whilst hydraulic gradient is an analogue for slope, modified by ice overburden pressure
(Schwanghart & Scherler, 2014). Several ribbed moraine theories indicate the importance of subglacial water's presence at the
bed of the FIS (Vérité et al., 2021; Ahokangas et al 2021; Möller, 2018, 2015; Peterson & Johnson, 2018; Peterson, Johnson
& Smith, 2017), whilst Shackleton et al. (2018) suggests that subglacial hydrology was highly responsive to minor changes in
ice conditions due to the low hydraulic gradient under the FIS. Our results reflect this somewhat, with hydraulic gradient being
much shallower than slope gradient for both ribbed moraine and non-ribbed moraine, whilst also showing hydraulic gradient
to have a wider "narrow band" of ribbed moraine features, indicating the additional factors in hydraulic gradient versus slope.
Thus, we note the importance of subglacial hydrological conditions in terms of ribbed moraine formation. This is, in addition,
aided by studies which demonstrate at least two of our areas of ribbed moraine formation to have large quantities of subglacial
water present during the existence of the FIS using geomorphological evidence of tunnel valleys, eskers and murtoo fields
(Shackleton et al., 2018, Voren & Plassen, 2008, Jansen et al., 2014, Dewald et al., 2022, Ottesen et al., 2020; Ahokangas et
al., 2021). These studies further suggest the importance of subglacial water, as rather than simply being a possibility based on
numerical modelling *if* water were present, they couple with Patton et al. (2017) identifying subglacial drainage through Europe
from areas such as Femundsmarka, along with factors mentioned in section 4.1.1, therefore meaning subglacial water may
play a part in the ribbed moraine's formation. Hence, ribbed moraines appear to form within a region with water presence, low
hydraulic gradient versus the average, and high hydraulic head versus the average.

Finally, we consider ice flow velocity as a key global predictor, showing a general concentration of ice flow for ribbed moraine
below 100 m a$^{-1}$, with some higher outliers in areas close to ice streams, or areas potentially formed outside of our 21 ka BP
timestep. This is consistent with ice flow in many prior studies, including modelling studies (Ely et al., 2023; Barchyn et al.,



2016), which show ribbed moraine to form rapidly, and in areas of relatively slow ice flow velocity. Where these conditions prevail, ribbed moraine are observed to form from instabilities in a basal till sheet (Möller & Dowling, 2015; Lindén, Möller

& Adrielsson, 2008; Möller, 2006), before being further modified into hummocks, drumlins and herringbone structures (Ely et al., 2023; Barchyn et al., 2016). Furthermore, one study (Putniņš & Henriksen, 2017) has suggested that ribbed moraines may form during the slow-down and speed-up of ice in the transition from large-scale driven ice flow to more regional ice flow, potentially at ice stream initiation. As a result, both from literary evidence and this study, we can conclude ribbed moraines to likely be a product of local relatively slow ice flow conditions. The short temporal formation period raises

difficulties for this study, however we are confident that with the data we have, 21 ka BP is the most accurate time period, especially due to the small differences between 21 ka BP and 14 ka BP within our data, and that 21 ka BP is close to the initiation of the FIS' deglaciation cycle.

Ice thickness and basal temperature appear to relate to elevation, but are not directly proportional to it. These values represent glacial status, and thus are representative as to whether the ice conditions have an influence on ribbed moraine formation.

From our total results, we can clearly see that both values have a similar issue to elevation. This is that there are several peaks which correspond to each of our subregions (FIG 4-8c), however they are not limited to a strict narrow band of values in a global sense, only regionally. Ice thickness shows the strongest global connection, but the ribbed moraine value distribution still encompasses ~80 % of the total. This could explain why hydraulic gradient is less distinct than slope as a global value, as the former contains information from the ice thickness value, specifically because it largely depends on ice surface slope, a

derivative of ice thickness. Basal temperature suffers a regional limitation, however, this is still not fully explained on a regional basis alone. This could be due to the coarse resolution of Patton's (2016; 2017) glaciation model at 10 km resolution, even with resampling to 250 m resolution in this study. In considering this, it may be possible that Shackleton et al.'s (2018) theory that subglacial conditions are strongly impacted by small variations ice conditions holds some weight in the formation of ribbed moraine. In this case, we would consider the base of the FIS during the formation of ribbed moraine to possibly be a

discontinuous patchwork of warm and cold based ice, fluctuating around the pressure melting point based upon whether the ice sheet base sits within a shallow basin or a convex ridge (Jansen et al., 2014).

Thus, from our results thus far we suggest the potential for a connected but discontinuous patchwork of high pressure, low gradient subglacial water flow to be a potential factor in ribbed moraine formation, drawing in the connection of each value as

discussed above.

## 3.2 Distribution in space

Here we will discuss in relation to subglacial hydrology – tunnel valleys, murtoos, eskers, canyons, sedimentary detail. We also consider the presence of ice streams as indicated by prior studies (Montelli et al., 2017; Ottesen et al., 2008) and the major fjords along the Norwegian coast, this is displayed in figure 1.



### 3.2.1  Subglacial hydrology

As literature suggests that both our South Norway under 900 m and North Norway study regions likely contained areas of subglacial hydrology during the presence of the FIS (Shackleton et al., 2018, Voren & Plassen, 2008, Jansen et al., 2014, Dewald et al., 2022, Ottesen et al., 2020; Ahokangas et al., 2021; Bjarnadóttir, Winsborrow & Andreassen., 2017), we are able to consider the presence of subglacial hydrology as a given within this study, and thus the importance of hydraulic head and hydraulic gradient is high. Some of these studies also identify hummock tracks and oblique ribbed moraine within tunnel valley systems and in proximity to subglacial water systems (Dewald et al., 2022; Ahokangas et al., 2021; Peterson & Johnson, 2017; Stroeven et al., 2016; Punkari, 1997). All of this evidence thus identifies a potential relationship between subglacial hydrology and the formation of ribbed moraine – both through geomorphological evidence in literature, and modelled evidence in this study.

In a further test to identify a more direct hydrological relationship to ribbed moraine, we subtract elevation adjusted for isostasy values from our hydraulic head values as a means of visualising only the influence on hydraulic head from ice overburden pressure (Kazmierczak et al., 2022). In doing this we produce figure 6 where we can see the hydrological portion of this metric varies over space, but has a clear increase when close to areas of known subglacial hydrology (Femundsmarka in south Norway, and Finnmarksvidda in north Norway). Additionally, outside of a small portion of ribbed moraine in the border regions where dataset edges cause issues with the output, we only see a few small series of ribbed moraine fields with sub-300 m.w.e. values. Hence, this could suggest that ribbed moraines tend to form where there are relatively high subglacial water pressures in comparison to areas without ribbed moraines, with a relatively low subglacial hydraulic gradient, when combining our existing results with this spatial analysis.



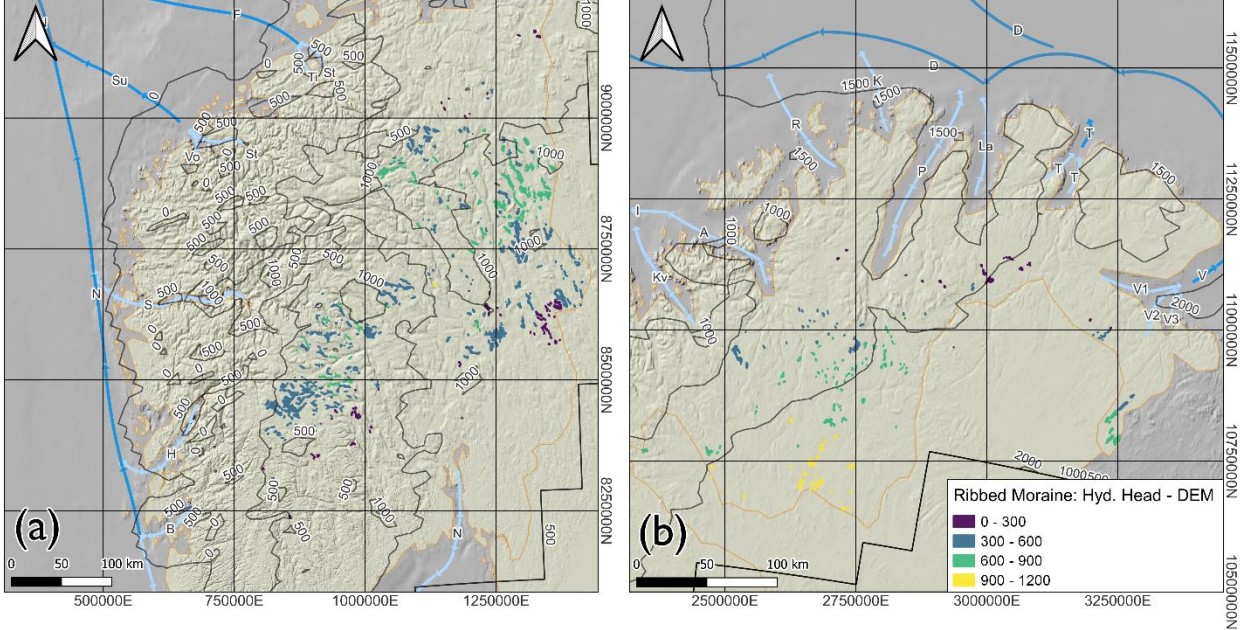

**Figure 6: Spatial distribution of ribbed moraine where colours represent values of hydraulic head – elevation, demonstrating the fraction of hydrological head produced by ice pressure, where (a) is southern Norway and (b) is northern Norway. Ice stream data from Ottesen et al., (2008) and basemap produced using Norwegian national data (Kartverket, 2016; ka BPrtverket, 2007).**

### 3.2.2 Ice Streams

An interesting factor is that many of the ribbed moraine are situated around the initiation points of major ice streams – for example the Femundsmarka ribbed moraine fields appear to be oriented towards the south, near the main Norwegian channel ice stream, whilst the Finnmarksvidda ribbed moraine are predominantly oriented towards the Porsangerfjorden, Tanafjorden and Varangerfjorden ice streams (Fig. 1). This is in line with theory from Putniņš & Henrikson (2017) that shows ribbed moraines as a feature formed during the activation and deactivation of ice streams, specifically in their onset regions. This relationship could partly explain the relationship between ice flow behaviour and ribbed moraine presence, but fits the hypotheses from Putniņš & Henrikson (2017) and Shackleton et al., (2018), that ribbed moraine can form in small patches, based on local scale variations in ice conditions, and that ribbed moraine may form during the slow down and speed up of ice flow during a phase transition. The location of ribbed moraines, upstream of the ice stream locations, would suggest that rather than forming under an ice stream, where features would be rapidly streamlined, they more likely form at the boundary between slow and fast-flowing ice, which could be interpreted as the margin of cold-based ice as suggested in prior studies (Stroeven et al., 2016; Hättestrand & Kleman 1999; Hättestrand, 1997; Sollid & Sørbel, 1994). Hence, the spatial distribution of ribbed moraines in relation to ice streams also provides some further information towards their formational conditions, namely reflecting the common understanding that they typically form close to the marginal area of cold and warm-based ice (Stroeven





et al., 2016). This could thus indicate that the marginal regions are less uniform than we currently understand, are form more of a patchwork of warm and cold based conditions driven by ice conditions (Shackleton et al., 2018).

### 3.3 Predominant conditions for ribbed moraine in literature and this inventory

Here we summarise our work and the literature, and discuss the conditions most commonly associated with ribbed moraine. These are divided regionally, and we conclude which conditions have the most predictive power based on their charts. We also combine with literature to discuss the (thus far) most common conditions and result with a common value set for ribbed moraine formation.

### 3.3.1 Relationships between ribbed moraines and modelled data

To summarise our findings, we identify that ribbed moraine form with a series of specific conditions measured at 21 ka BP. Specifically, ribbed moraines are present in areas with higher hydraulic head, low hydraulic gradient, low ice flow velocity, and low slope and curvature. In terms of landscape geomorphology, ribbed moraines are present in areas of between $0 – 6.7$ m m$^{-1}$ slope (0 - 8.5 ° gradient) and between $-8.2 \times 10^{-3} – 6.5 \times 10^{-3}$ m m$^{-2}$ curvature. Thus, in terms of geomorphology ribbed moraine form in a narrow band of low curvature and low terrain slope, suggesting valley, basin or topographic depressions. Due to the presence of ribbed moraine in both negative and positive curvature environments, it is also possible to find ribbed moraine on convex hillslopes of low grade, however they have a higher limit of negative curvature (concavity) versus positive curvature (convexity).

Hydrologically, ribbed moraines are present in regions where hydraulic head has been modelled between 1001 m – 1865 m, whereas hydraulic gradient is gradual. Hydraulic head shows a less uniform distribution versus the geomorphological factors, with a peak in values at 1577 m, and a minimum value above the mean of the non-ribbed moraine values. On the other hand, hydraulic gradient, like geomorphological slope, shows a lower mean and median value than the non-ribbed moraine data. Thus, we note ribbed moraines are most commonly present in areas of high hydraulic head, and by association, water pressure, but low hydraulic gradient. This could indicate a high-pressure subglacial drainage system played a part in their formation, if temperate conditions were present.

When considering ice conditions and dynamics there is less prediction power on a global scale, with only ice velocity playing a strong role in its value as a predictor due to its narrow band (with outliers) for ribbed moraines versus areas without ribbed moraines. Ice velocity in our study for ribbed moraine ranges from $0 – 256$ m a$^{-1}$, with a mean and median at 65 m a$^{-1}$ and 45 m a$^{-1}$ respectively. This shows that whilst there is a large spread of higher velocities, the averages (Fig. 5c) are predominantly pooled below 100 m a$^{-1}$. As such, we can conclude we find ribbed moraines to form in low-velocity regions, where ice is indeed flowing, but at rates of sub-100 m a$^{-1}$. In summary, we determine ribbed moraines to be present in areas of very low slope and curvature, high hydraulic head, but low hydraulic gradient, and low ice velocity. These characteristics are most common with small depressions and basins in the landscape, where ice is relatively thick. Hence, spatially we can suggest ribbed moraines do not form at the margins or the core of an ice sheet, but at a zone in between, where basal conditions are





transitioning from cold to warm (Putniņš & Henriksen, 2017; Trommelen et al., 2014). This can be further explored and validated when combined with literature.

### 3.3.2 Comparing our findings to literature

When combining our findings with literary evidence for ribbed moraine formational conditions, we find several commonalities. Our geomorphological conditions are reflected in several studies (Sommerkorn, 2020; Dunlop & Clark, 2006; Barnes et al.,
2024), while our modelled flow velocities are consistent with recent modelling studies (Ely et al., 2023; Barchyn et al., 2016). As such it is very likely that ribbed moraine form within ice which is flowing, but at a relatively slow rate – a theory which is further reflected by Putniņš and Henriksen (2017). Hydrologically, our results seems to be supplemented by several studies which indicate there was subglacial water flowing in the regions where we find ribbed moraines today (Shackleton et al., 2018, Voren & Plassen, 2008, Jansen et al., 2014, Dewald et al., 2022, Ottesen et al., 2020; Ahokangas et al., 2021; Bjarnadóttir,
Winsborrow & Andreassen, 2017), but further evidence for ribbed moraine having a relationship with subglacial hydrological conditions is also put forward. Dewald et al., (2022) outline the relationship between subglacial hummocks and ribs within tunnel valleys and water flow corridors under the former FIS, whilst Lindén, Möller and Adrielsson (2008) find through sedimentological analysis that ribbed moraine are potentially formed by sediment stacking, followed by hydrological cavity infill, a theory further built upon by Fowler and Chapwanya (2014) who discuss the importance of subglacial film-flow in
landform formation. In coupling these with our study's findings, we note that ribbed moraine are likely related to a subglacial cavity flow system, with high subglacial water pressures (represented by our high hydraulic head values), and low hydraulic gradient. This could thus be further built upon by the former cold to warm based ice divide being a discontinuous patchwork (Shackleton et al., 2018), allowing for the existence of ribbed moraines in proximity to features requiring cold based ice for either preservation or formation.

### 595 3.3.3 Addressing theories of formation

When considering our results, and the modelled formational conditions of ribbed moraine - primarily from Ely et al., (2023) and Barchyn et al. (2016), we consider it possible to narrow down the list of formational theories of ribbed moraine. Trommelen, Ross & Ismail, (2014) outlines the conditions required for many of the predominant theories of ribbed moraine formation, further allowing for this. Our results in summary show that ribbed moraine form in topographic depressions, where
ice flow is relatively slow, there are high subglacial pressures and low subglacial gradient. Literature has further determined the presence of hydrological systems within the region (Shackleton et al., 2018, Voren & Plassen, 2008, Jansen et al., 2014, Dewald et al., 2022, Ottesen et al., 2020; Ahokangas et al., 2021; Bjarnadóttir, Winsborrow & Andreassen, 2017). Hence, if we take the model at face value, we can address theories that require only cold-based ice (Lundqvist, 1997; Hättestrand & Kleman, 1999; Sarala, 2006), and state that these are likely formational processes. Yet, if we consider the uncertainty within
the glaciation model we use, there are potential patches of subglacial water under the FIS, particularly in the areas where we find ribbed moraines due to greater than average ice thicknesses. As a result, it is possible to argue against the validity of



theories requiring simply cold-based ice if we accept that water may be present in topographic depressions, where ribbed moraines are commonly found today. As a result, our acceptance of likely uncertainties within the ice model (Patton et al., 2016; 2017) at smaller scales than the 10 km grid cells, means that theories which suggest a patchwork or transitional zone of cold to temperate ice (Shackleton et al., 2018) may hold more weight than simple "solely cold-based" models.

Furthermore, when considering recent modelling studies (Ely et al., 2023; Barchyn et al., 2016), it becomes clear that ribbed moraines tend to form in a short period of time, and are then potentially preserved, making the transience of the cold-warm patchwork both spatial and temporal for the preservation of ribbed moraine. Our results also indicate the possible subglacial water presence to be both high pressure, but low gradient, thus suggesting cavity or film flow, which is in opposition to early versions of the "shear-stacking" hypothesis (Sollid & Sørbel, 1994; Shaw, 1979), which suggest basal sediments are sheared and stacked by freezing to the base of the glacier, indicating cold based ice. We also consider the opposite to be untrue – that there were large amounts of subglacial flow, causing ribbed moraine to be formed in major outburst flooding (Fisher & Shaw, 1992).

While our results do not point to one unifying theory in formation, we do not have any direct contradictions of the instability theory (Ely et al., 2023; Barchyn et al., 2016; Fowler & Chapwanya, 2014; Chapwanya, Clark & Fowler, 2011; Dunlop, Clark & Hindmarsh, 2008), nor more modern interpretations of the shear and stack theory (Lindén, Möller & Adrielsson., 2008), nor do we find any major contradictions to Trommelen, Ross & Ismail's (2014) "sticky spot" theory. Instead, we find components of all three theoretical families to be compelling when combined with our results. For example, the patchwork of subglacial temperatures connects strongly with Trommelen, Ross & Ismail, (2014) when considering Shackleton et al., (2018), whilst cavity flow theories reflect the cavity infill discussed in Lindén, Möller and Adrielsson (2008), and our measurements of flow velocity reflect the modelling studies by Ely et al., (2023) and Barchyn et al., (2016). Furthermore, the short timescales put forward in Putniņš and Henriksson (2017) are a compelling addition to these theories. As such, it may be possible to reflect on Möller and Dowling's (2018) study, in which they conclude that ribbed moraine may be a product of "equifinality", where multiple processes may end up with the same resultant landform. This can already be seen in part through Ely et al., (2023), where simply the presence of bed instabilities results in ribbed moraine formation in all simulations. Thus, from this study we are able to narrow down ribbed moraine formation to a more specific set of values than before, but we must stop short of a unifying theory of formation, and rather address the potential of an equifinal production method.

## 4    Conclusions

In this study, we used a nationwide ribbed moraine inventory produced using (Barnes et al., 2024), and from this we identified the constraining global datasets of slope, curvature, hydraulic head, hydraulic gradient and ice flow velocity, for a reasonable timestep (21 ka BP). By identifying these values, we have addressed the prevailing ribbed moraine formational theories, applying both our findings and the insight of recent modelling and hydrological analysis studies of Fennoscandia (Ely et al., 2023; Shackleton et al., 2018, Voren & Plassen, 2008, Jansen et al., 2014, Dewald et al., 2022, Ottesen et al., 2020; Ahokangas





et al., 2021; Bjarnadóttir, Winsborrow & Andreassen, 2017, Barchyn et al., 2016). However, even with these new findings we
could not confirm the formational process of ribbed moraine, but have constrained values so as to narrow down the potential
formational theories from the current wide-ranging set. As such, we note our main conclusions as:

    a.   In keeping with prior literature (Dunlop & Clark, 2006), ribbed moraine typically form in flat, low gradient and low
        curvature depressions in the landscape. In these locations, hydrological conditions have been modelled as high
hydraulic potential and thus pressure, but low hydraulic gradient, furthermore we note that in these regions ice velocity
        is generally lower than average.

    b.   Factors such as elevation and ice thickness are too spatially variable to be valuable global predictors for ribbed
        moraine presence, instead being strong local predictors on local or regional scales. Furthermore, basal temperature is
        too coarsely derived to be a strong global predictor. In this study we find north Norway, and two subsections of south
Norway to be distinct regions for these three values.

    c.   Despite the low resolution of model data, we gather strong evidence for the conditions outlined in (a) through
        interpolation and reference to literature. Several prior works (Shackleton et al., 2018; Trommelen, Ross & Ismail,
        2014) suggest a patchwork of warm and cold-based glacial conditions during the FIS' presence, and thus we consider
        similar conditions around ribbed moraine formation. This can explain the proximity of ribbed moraine to streamlined
and other non-transverse landforms.

    d.   Our results effectively suggest the validity of the natural instability (Ely et al., 2023) and modern shear and stacking
        (Lindén, Möller & Adrielsson, 2008) theories whilst providing evidence against the validity of others. For example,
        the presence of hydrology within the regions of ribbed moraine formation suggest that fracture and extend cold-based
        processes cannot occur, and thus ribbed moraine cannot form in this way. Furthermore, our findings coupled with
modelling studies (Ely et al., 2023; Barchyn et al., 2016) disagrees with the slow, repeated modification of pre-
        existing subglacial landforms, as ribbed moraines are modelled to form over a short timescale from bed instability.

    e.   Ribbed moraine potentially form in warm spots at the bed of the former FIS, with high-pressure cavity or film-flow
        hydrology influencing their formation. These could be interpreted as sticky spots within a wider warm bed, where
        large areas of cavity hydrology slow down ice flow (Trommelen, Ross & Ismail, 2014), or conversely as warm spots
within a cold bed, where cavity infill occurs as in Lindén, Möller and Adrielsson, (2008). Yet with current ice models,
        we are unable to determine whether this is the case due to resolution limitations. However, it is highly likely that in
        either case, bed instabilities play a significant role given their role in recent modelling studies.

    f.   We conclude the likelihood of an "equifinality" explanation for ribbed moraine formation, where there is potential
        for both series of processes to result in similar end formats (Möller & Dowling, 2018). In such a case, the conditions
would vary within our data range, but the result would remain the same.



## 5 Author Contribution

For this work, TJB carried out the majority of research and writing. TVS carried out discussion and project planning, along
with supervision duties and manuscript editing/ reviewing. KSL contributed similarly to TVIS, through discussion and project
planning, and supervision with manuscript reviewing. Finally, LSS contributed to the data processing, scripting and manuscript
reviewing. In total TJB prepared the manuscript, but all co-authors contributed with review comments.

## 6 Competing Interests

The authors declare that they have no conflict of interest in this manuscript.

## 7 Acknowledgements

We acknowledge Dr. H. Patton of Tromsø University's contribution through production of his ice sheet model and contribution
of model outputs to this work, they were greatly appreciated and were essential to carrying out this project. We further
acknowledge Dr. E. M. Lund of the Norwegian Geological Survey, and Dr. H. Åkesson of the University of Oslo for minor
proofreading and discussion within the field of this work.

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
