# Peer review of "Formation Conditions of Ribbed Moraine in Norway: A Distribution Analysis and Ribbed Moraine Inventory"

_EGUsphere, 2025_

## Author Comment (AC1)

**General response to reviewers:**

We thank the reviewers for their contributions. The feedback provided have been of great value for improving this manuscript, and we hope that our revisions address the comments in a satisfactory way. Below, we list the comments (bold face) and reply (normal face) to them point by point. The numbers in italic face refer to original line numbers from the version read by the reviewers. Further, responses in red text are large edits based on review comments, whilst responses with green text are relevant sections of the manuscript to the response, regardless of edits.

**Response to reviewer 1:**

**31: And Iceland, for the first time described by Nína Aradóttir et al. 2024**

*Thanks for the reference, this has been amended and added to the reference list.*

**52: There appears some confusion here, for example Linden et al 2008 subscribe to the 'polygenetic' school of thought.**

*Apologies for the confusion, we understand that Lindén et al. (2008) state their agreement with polygeneticism of ribbed moraines – this has been addressed in the text.*

**55: And generally linked to compressive ice flow?**

*Added a few words to this to specify "often compressive flow", and a citation to "Stokes et al., 2008" – it is not in all cases, but in many that they are associated with compressive ice flow, line 56, S. 1)*

**90: Check numbering of sections**

*The numbering issue should have been addressed now.*

**100: I think you should briefly explain the methodology from the 2024 paper e.g., it's not clear that it's a machine-learning algorithm.**

*We have now added a brief couple of sentences explaining that this methodology was machine learning based and how it works in basic terms. This is now present in lines 101 – 105, S. 2.1.*

**105: If there exists a national database of landforms from Kartverket, I don't immediately follow what the role of the machine-learning approach is? Are there gaps in the Kartverket database? Does your algorithm find ribbed moraines missed by Kartverket? What is the detection accuracy in this study? It sounds like the algorithm could be a useful tool to apply elsewhere too, so some additional information on its performance for creating an inventory at this scale would be relevant for future users.**

*We note that our citation to the dataset needs updating to NGU, 2016, but the point remains the same here. The database from NGU is incomplete, and based on old approaches and data that do not hold up in terms of accuracy versus the modern high-resolution elevation model data. The method in Barnes et al., 2024 identified both ribbed moraines that were not present in the NGU data and also was specific in ribbed moraine location, rather than marking a general area from aerial imagery. Additional information on its performance is given in Barnes et al. (2024).*

*To address this comment, we update section 2.1, adding some sentences on why we use both approaches – specifically as they complement one another, as the old data is not "completely*

*useless", but rather, lower resolution versus modern approaches, whereas the modern data still misses things that may have been picked up by older approaches. Lines 121 – 134, S. 2.1.*

*(Barnes & Filhol, 2023). This methodology makes use of a K-means algorithm to group elevation data based on its morphological characteristics. The segmented data is divided into ribbed moraines and other components based on the contents of each group, or "cluster", of data in the output at an accuracy rate of 65 – 75%.*

*This ensures that we work with high-quality and accurate landform polygons for Norway, as the data from the Norwegian national databases proved incomplete and of low accuracy, being based on old methodologies and data (Sollid & Torp, 1984; Hättestrand & Kleman, 1999). The K-means methodology from Barnes et al., (2024) proved this approach valuable by detecting ribbed moraines not previously mapped in Norway, and being iterable across areas that detailed superficial geology maps were not available. As a means of ensuring maximum coverage and accuracy, we merged the automated data with nationally mapped landforms (Kartverket, 2021), allowing both datasets to complement the others' shortfalls. For example, this allows the datasets to supplement one another where the old data is incomplete, and because results from the automated approach reach 65-75% accuracy (Barnes et al., 2024). Hence, both methods complement one another, as it has been shown that manually delineated features in spatial data often differ greatly from automated mapping (Corr, 2024).*

**110: Will this dataset be made available? I think it would be a useful resource for any future workers wanting to build on this work.**

*There were not plans to make this dataset available, however the algorithm is made available as a companion to Barnes et al., (2024), we have added this as a citation. We have cited the algorithm as (Barnes & Filhol, 2023). We have now uploaded the ribbed moraine field dataset to Zenodo, and added a data availability statement including this. This is cited as Barnes, (2025).*

**130: UiT model is more commonly used reference rather than Patton model.**

*Noted and adjusted!*

**234: Maybe the Shackleton paper is not the best citation here - there is a lot of literature on the subglacial thermal organisation of ice sheets where this 'patchy' ice resonates e.g., Kleman & Glasser 2007.**

*On re-reading, this makes sense, we have addressed this, another possibility is the Stokes et al., 2008 paper on ribbed moraines. Lines 273 – 276 S. 2.4.*

**256: Define what a murtoo is as it is probably not widely known.**

*Added a short sentence outlining the current understanding of murtoos (lines 297 – 299, S. 3.1, we also cite Ahokangas et al., 2021 and Ojala et al., 2021).*

**341: While reading the results it may be worth considering using acronyms for the data subsets e.g., southern Norway under 900 m elevation = SN<900, just to make it a bit easier to digest.**

*Adjusted, now when referring to study subsections, NN is North Norway, SN<900 is south Norway under 900 m, and SN>900 is south Norway above 900 m.*

**408: I think this is an important section on the limitations regarding the comparison with model data. But I think there is also a temporal aspect to consider. For example, you are**

comparing data to a snapshot in deglaciation, and quite early too. It is not realistic to assume all ribbed moraine in Norway are contemporaneous so the glaciological conditions examined are likely to have some inherent uncertainty linked to temporal variations. From other modelling it has been shown that basal conditions can vary significantly over short timescales e.g., Patton 2022. I think you briefly touch upon this on line 227, and it's an interesting point that you see similar trends at 14 ka even with a reduced dataset.

*Thanks for this, we have added a paragraph at the start of 4.1 addressing the issues brought up in this comment, explaining our reasoning for sticking with the 21 ka BP snapshot, whilst also accepting that it is not realistic to assume that all ribbed moraines in Norway formed contemporaneously. Lines 455 – 462, S. 4.1.*

*Our results are modelled at 21 ka BP due to the flow orientation of the FIS and the orientation of mapped ribbed moraines matching one another at this snapshot. As such, our findings are true for ribbed moraines that are formed at the 21 ka BP snapshot, and to a lesser extent, the 14 ka BP snapshot due to similar, but lesser trends in ice flow orientation. Hence, the values derived for ribbed moraine formation may not reflect the exact conditions at the point of formation of these ribbed moraines, especially as basal conditions have been shown in models to vary significantly over short timescales (Patton et al., 2022). However, it is difficult to fully determine the point of formation of ribbed moraines, and as such, we focus on the 21 ka BP snapshot due to the flow direction commonalities in the model data and mapped features, whilst also accepting that it is not realistic to assume all ribbed moraines in Norway did not form contemporaneously.*

**424: To clarify, the figure referred to here shows the potential routing of subglacial drainage, even if no subglacial water was present (check also line 467). Otherwise, I agree that subgrid anomalies are to be expected, particularly in areas of high relief.**

*Thanks for the added clarification here, we found this figure particularly interesting, as we had been trying to carry out similar hydrological flow routing on a narrow scale, to a lack of great success. We have clarified this in line 479, S. 4.1.*

**525: Misformatted citation.**

*Thanks, resolved!*

**578: I'm not sure that this section is necessary as it repeats what you have already discussed earlier in the discussion.**

*Understood, we have cut this section, the initial idea with this was to tie things together, but we agree that repetition should be minimised.*

**600: This should probably be more specific on what pressure and gradients - hydraulic.**

*Changed to "high subglacial hydraulic pressures and low subglacial hydraulic gradient". Line 656, S. 4.4.2*

**609: And evolving glaciological conditions through deglaciation.**

*Added to lines 665 – 666, S. 4.4.2.*

**616: Why is the flood hypothesis dismissed?**

*We used too strong language, rather than "untrue" we have changed to "unlikely". We consider this unlikely because the hydrological system doesn't appear to be high enough capacity to*

*warrant this based on the model's data at least. Although, it is hard to resolve of course. We have added some words on this. Lines 683 – 688. S. 4.4.2.*

**631: It would be useful to say what is needed to move this forward (future directions) e.g., higher-resolution model outputs, a time-transgressive analysis of glacial parameters, examination of other domains...**

*We have added a short paragraph on this, primarily addressing model resolution, temporal factors and a need for some more comprehensive sedimentological studies: "To make further steps in understanding ribbed moraines, we would need a glacier model of higher temporal and spatial resolution, capable of resolving sub-valley scale changes in ice conditions. Such a model could be used to analyse temporal variability of conditions on the scale of several ribbed moraine fields, allowing for new insights into specific potential times of formation, rather than a general timestep estimation as in this study. Additionally, comprehensive sedimentological examination of ribbed moraines would likely provide strong insights into their formation, as sediment composition analysis in former works (e.g. Lindén, Möller & Adrielsson, (2008)) has been limited to single, or small areas of bedforms." Lines 704 – 709, S. 4.4.2.*

*To make further steps in understanding ribbed moraines, we would need a glacier model of higher temporal and spatial resolution, capable of resolving sub-valley scale changes in ice conditions. Such a model could be used to analyse temporal variability of conditions on the scale of several ribbed moraine fields, allowing for new insights into specific potential times of formation, rather than a general timestep estimation as in this study. Additionally, comprehensive sedimentological examination of ribbed moraines would likely provide strong insights into their formation, as sediment composition analysis in former works (e.g. Lindén, Möller & Adrielsson, (2008)) has been limited to single, or small areas of bedforms.*

**Response to Reviewer 2:**

**This paper presents some interesting observations of subglacial rib occurrence across Norway. However, I find the scope of this paper, and the insights it provides to be rather limited. The authors derive no real insight into rib formation, and the dataset is not particularly useful for constraining rib formation in models.**

*Thank you for considering this paper, we approach the concept of "rib formation" by addressing conditions, rather than processes in this paper. The scope and goal of the paper is limited by design, and was not to specifically address theories in great detail, but rather to provide additional information or back-up pre-existing ideas using ice sheet model data. This was primarily for the purpose of having a geomorphological and ice condition-based understanding of rib formation, something which is lacking versus the present physics and numerically derived formation models. In summary, the targets of this paper were to find connections and relationships between ribs and ice/water/geo conditions, rather than to determine specific processes.*

***Major comments***

**The distribution of ribs and their characteristics was documented by Hättestrand and Kleman in 1999, - how does the database here (Figure 1) improve on this?**

*The database in this study builds on Hättestrand & Kleman (1999) by automatically mapping ribs using elevation data, whilst Hättestrand & Kleman (1999) (along with Sollid & Torp (1984) for Norway) used primarily optical imagery. The method itself was illustrated in Barnes et al., 2024, which showed the automatic machine learning approach detected ribs not previously mapped in*

existing geomorphological data. We address this in more detail in reference to the comment on line 105 by reviewer 1. To summarise it simply, the inventory used in this work combines old and new data, and thus achieves a greater extent of detail and accuracy with both datasets complementing one another. Lines 125 – 134, S. 2.1.

*This ensures that we work with high-quality and accurate landform polygons for Norway, as the data from the Norwegian national databases proved incomplete and of low accuracy, being based on old methodologies and data (Sollid & Torp, 1984; Hättestrand & Kleman, 1999). The K-means methodology from Barnes et al., (2024) proved this approach valuable by detecting ribbed moraines not previously mapped in Norway, and being iterable across areas that detailed superficial geology maps were not available. As a means of ensuring maximum coverage and accuracy, we merged the automated data with nationally mapped landforms (Kartverket, 2021), allowing both datasets to complement the others' shortfalls. For example, this allows the datasets to supplement one another where the old data is incomplete, and because results from the automated approach reach 65-75% accuracy (Barnes et al., 2024). Hence, both methods complement one another, as it has been shown that manually delineated features in spatial data often differ greatly from automated mapping (Corr, 2024).*

**I think throughout, the authors set up a straw man, a false dichotomy, between equifinality and equicausality. This conflict has been resolved and shown to be of little use by Fowler (2018), who shows how many models account for a finite amount of physics but infinite amounts of potential history through "equimorphology" essentially resolving this debate.**

*Thank you for your comment on this. We acknowledge that through the paper by replacing "equifinality" with "equimorphology" (Fowler, 2018) and have changed references to this throughout. Equimorphology resonates with the ideas we conclude on. Now, references to equifinality are "equimorphology" as per this comment, this was primarily an issue of terminology. We have also included citations to Fowler (2018) throughout where this reference is necessary.*

**Furthermore, I think the term "ribbed moraine" is misleading, with many now using "subglacial ribs" as they are not moraines in the traditional end of a glacier sense.**

*We believe that within the field of geomorphology, moraines refer to any unconsolidated debris deposited by a glacier, this is backed up by e.g. Goudie, (2004); Schomacker (2014). This, therefore, identifies moraine as more than only terminal moraines, but rather, a family of related features including basal and melt-out moraine. Whilst "ribbed moraines" in some cases have been shown to contain some sorted material, indicative of glaciofluvial processes, or at least, the involvement of subglacial water in their formation, within geomorphology this term is still commonly used. Further, "ribbed moraines" is used by recent literature in the form of Kamleitner et al. (2023), Aradóttir et al. (2024) and Ploeg & Stroeven (2025). With this backing we continue using the term "ribbed moraines".*

**My final point is that the authors use a single time-slice from a model for understanding rib formation. However, we have absolutely no idea whether these conditions are those in which ribs formed. Indeed, I would expect much of the area studied to be cold-based at the last glacial maximum, and thus no subglacial movement would have occurred.**

*Thanks for this comment, we address the time slice issue in section 4.1, where we have added a paragraph in reference to reviewer 1's point for line 408 (Lines 455 – 462, S. 4.1; Lines 256 – 261, S. 2.4). Furthermore, in the paragraph following there is reasoning as to why there may be patches*

*of warm and cold-based ice, specifically due to the model being "off" in elevation by values of up to and over 1000 m due to its spatial resolution. We address this in more detail in the line-by-line comments below.*

*Our results are modelled at 21 ka BP due to the flow orientation of the FIS and the orientation of mapped ribbed moraines matching one another at this snapshot. As such, our findings are true for ribbed moraines that are formed at the 21 ka BP snapshot, and to a lesser extent, the 14 ka BP snapshot due to similar, but lesser trends in ice flow orientation. Hence, the values derived for ribbed moraine formation may not reflect the exact conditions at the point of formation of these ribbed moraines, especially as basal conditions have been shown in models to vary significantly over short timescales (Patton et al., 2022). However, it is difficult to fully determine the point of formation of ribbed moraines, and as such, we focus on the 21 ka BP snapshot due to the flow direction commonalities in the model data and mapped features, whilst also accepting that it is not realistic to assume all ribbed moraines in Norway did not form contemporaneously.*

*To do this we determined most likely ice flow orientations for all ribbed moraine fields by calculating the angle transverse to landform orientation, and calculating the mean angle for each moraine field. We then compared these values to modelled ice flow orientation from Patton et al., (2016; 2017). This comparison was mode in timesteps every 1000 years, from 25 ka BP to 10 ka BP, resulting in 14 ka BP and 25 ka BP being two timesteps oriented most in line with mapped ribbed moraines.*

**This approach is therefore fundamentally flawed, and a better model-data comparison is required. Perhaps deriving flow direction from the ribs and the model, and looking at conditions when the flow direction and the ribs align with with each other might be a good start.**

*We derive flow direction from both the ribs and the model in this study, and then compare where they align most closely with one another – this ended up being at 21 ka BP and 14 ka BP to a lesser extent. This is now addressed in the new paragraph at the beginning of section 4.1, as in reference to line 408 from reviewer 1. We have also added a short sentence at the start of section 2.4 to aid in understanding. Lines 455 – 462, S. 4.1.*

*Our results are modelled at 21 ka BP due to the flow orientation of the FIS and the orientation of mapped ribbed moraines matching one another at this snapshot. As such, our findings are true for ribbed moraines that are formed at the 21 ka BP snapshot, and to a lesser extent, the 14 ka BP snapshot due to similar, but lesser trends in ice flow orientation. Hence, the values derived for ribbed moraine formation may not reflect the exact conditions at the point of formation of these ribbed moraines, especially as basal conditions have been shown in models to vary significantly over short timescales (Patton et al., 2022). However, it is difficult to fully determine the point of formation of ribbed moraines, and as such, we focus on the 21 ka BP snapshot due to the flow direction commonalities in the model data and mapped features, whilst also accepting that it is not realistic to assume all ribbed moraines in Norway did not form contemporaneously.*

**Minor comments**

**Abstract: Poorly written. essential says "we didn't find any relationships." Also, reflects the false fight between equifinality and equicausality mentioned above.**

*We have rewritten the abstract to clarify our findings, including stressing our support of the "equimorphology" concept, due to our previous conflation of equifinality / equicausality. Lines 6 – 23.*

*Ribbed moraines are common landforms in regions formerly glaciated by the Fennoscandian, British and Laurentide ice sheets. However, their process of formation is disputed, as past subglacial conditions are hard to reconstruct. In this study, we investigate possible formational conditions using a comprehensive ribbed moraine inventory of mainland Norway, mapped topographic information, and modelled glacial and hydrological information.*

*We first produce a high resolution (10 m) ribbed moraine dataset for the entire mainland Norway using a machine learning algorithm. This dataset is used to identify common characteristics for ribbed moraines in nine variables: (a) topographic (elevation, slope and curvature), (b) glacial (basal temperature, ice thickness and ice velocity), and (c) hydrological (flow accumulation, hydraulic head and hydraulic gradient). These variables are evaluated at 21 ka BP. Values for areas of ribbed moraines are compared to an equal area where ribbed moraines are not present.*

*Our findings show that (a) ribbed moraines typically form in low gradient and low curvature depressions with a low hydraulic gradient, (b) hydraulic head, hydraulic gradient and ice velocity are globally important for ribbed moraine formation, while factors such as elevation and ice thickness are too spatially variable for a wide-scale link to be drawn, but they may be important locally, (c) ribbed moraines are present in areas where ice flow was relatively slow, (d) their occurrence in areas of high hydraulic head, low hydraulic gradient and low ice velocity suggests formation in transitional areas between slow and fast ice flow, which may resemble a "patchwork" of slippery and sticky spots of high and low frictional resistance.*

*While the found relationships are robust, they do not confirm a specific formative mechanism. We therefore suggest that ribbed moraines may result from multiple processes operating under similar boundary conditions, supporting an "equimorphology" concept explanation for their formation.*

**L44: See comment above about how this dichotomy is false.**

*We have addressed this by changing references to "equifinality" to now say "equimorphology". We have also added reference to Fowler (2018).*

**L90: Something wrong with the numbering (two section 1s).**

*Section numbering has been fixed.*

**L72: Misrepresentation of Ely et al. (2023) - rib formation is it is not linked to ice velocity, but effective pressure at the base. The point of this paper, and indeed the models beforehand from Fannon, Fowler and Hindmarsh, is that an instability occurs which makes linking the formation to ice flow conditions intractable. Thus, the attempt to do so in this paper is perhaps a bit misguided. The Barchyn paper has also been discredited as being physically unrealistic (see Fowler, 2018). I suggest rewriting after reading these papers properly.**

*Thanks for bringing our attention to this, we note the mistake of misinterpreting Ely et al. (2023), and any discussions of relationships to ice flow velocity and replace these points with references to studies that suggest a velocity connection to ribbed moraines, for example, Vérité et al. (2023), Stokes et al. (2008), Finlayson & Bradwell (2008). We also address the concerns about Barchyn et al., 2016 and the physical issues with the model. We have also added further context which outlines the current state of the art in terms of relating ice flow velocity to bedform formation, including reference to older suggestions thereof (e.g. Stokes et al., 2008; Finlayson & Bradwell, 2008), and more recent studies addressing such (e.g. Vérité et al., 2023). We hope this sufficiently addresses the concerns raised here through the addition of several lines to the manuscript, copied below in red. Lines 82 – 89, S. 2.1, minor changes throughout.*

*However, since the used model is physically unrealistic (Fowler, 2018), this result should be interpreted with caution. Whilst Barchyn et al.'s (2016) conclusions are thus, difficult to lend weight to, several other studies into ribbed moraines suggest that their formation is related in some way to ice flow, whether in the form of "slow-down, speed-up" (Dunlop & Clark, 2006; Putniņš & Henriksen, 2017) ice movement, or compressional ice flow (Finlayson & Bradwell, 2008; Stokes et al., 2008). This paper, therefore, aims to investigate, amongst other factors, whether ice flow conditions are a possible factor in bedform formation, as instability-based models such as Ely et al., (2023) suggest otherwise, while other studies suggest that velocity has a part to play in the rate of formation (Dunlop, Clark & Hindmarsh, 2008; Vérité et al., 2023).*

**L94: How realistic is it to interpolate ice sheet model conditions to this scale? How was this interpolated?**

*We have addressed this in place – while interpolating to 250 m does allow for analysis of values at a smaller scale, particularly where data involves use of the national DEM (hydrological and geomorphological data), this does not specifically detract from the model. Otherwise, interpolating down from 10 km and resampling up to 250 m from 50 m was used as a means of homogenizing data resolution, while avoiding detail loss in the other datasets. Lines 110 – 112, S. 2.*

*We chose 250 m so as to not require upscaling of morphological and hydrological data to scales wider than valleys, as this would cause a loss of geomorphological detail. This scale also allowed for interpretation of data within moraine fields, as these often reach over 250 m diameter.*

**143: Now you say you do every timestep - why didn't you analyse every timestep? 2.3.3**

*We have addressed this issue, during this study we did test every timestep, we have made this clearer. Lines 170 – 171, S. 2.2.1; Lines 256 – 261, S. 2.4; Lines 267 – 270, S. 2.4.*

*Using the glaciation model described in Patton et al. (2016, 2017), we derive a series of glaciological conditions for every 1 ka BP timestep between 11 ka BP and 25 ka BP. We test these timesteps versus ice flow orientation in the model and in ribbed moraine fields to determine which match up most closely.*

*To do this we determined most likely ice flow orientations for all ribbed moraine fields by calculating the angle transverse to landform orientation, and calculating the mean angle for each moraine field. We then compared these values to modelled ice flow orientation from Patton et al., (2016; 2017). This comparison was mode in timesteps every 1000 years, from 25 ka BP to 10 ka BP, resulting in 14 ka BP and 25 ka BP being two timesteps oriented most in line with mapped ribbed moraines. In comparing the timesteps of 14 and 25 ka BP.*

*This is not to say that all ribbed moraines mapped in this study are from the same timestep, as they likely formed at different times throughout deglaciation. Instead, we use the 21 ka BP timestep as within the data we have, this is the most commonly matching point in time for ribbed moraine field orientation, and ice flow orientation.*

**Section 1.3.1 - I don't get why you subdivide at all. It doesn't appear logical to me as stated. Surely this introduces bias into your analysis?**

*We have addressed this with a sentence to make it clearer: "This was done so as to determine whether patterns originating from our analysis were local scale (to a subdivision) or national scale (common across all fields in Norway)." Lines 228 – 229, S. 2.3.1.*

**L228 - As mentioned above, I don't think it is reasonable at all to use the 21 ka BP period solely. Why only use these two times when comparing? I am a bit baffled. Of course some ribs will be deglaciated by the latter time slices, but that doesn't mean that the ice flow conditions at 14 for those that were covered don't relate to the formation time. This reads as "we picked 21ka because it was easy, not because it was scientifically correct"**

*This is now section 2.4. We have addressed this in text, with several additions explaining that we began with each timestep, compared ribbed moraine orientation to ice flow direction, and settled on 21 ka over 14 ka after further testing. We settled on 21 ka over 14 ka as 14 ka depicted ice margins that would place many ribbed moraine fields beyond the margins of the ice sheet. Further, "21 ka BP" was the timestep which matched up the most with ribbed moraine orientation and model orientation. We acknowledge that ribbed moraines do not all form at exactly the same time period, but it is also impossible to resolve exactly when every field originated. Thus, we use 21 ka BP as our "best guess", as it is not unlikely that many of the conditions present during field formation persisted in some way, particularly in reference to compressional sticky spots (Stokes et al., 2008). The main goal here was to explore or investigate the subglacial conditions at this point and see if there are similarities at this full scale. Trommelen et al. (2014) shows that it is debated as to the conditions of formation, and they could be warm, cold, wet or dry. This must not be misunderstood as we think that everything formed at 21ka. Lines 256 – 261, S. 2.4; Lines 267 – 270, S. 2.4; Lines 273 – 276; S. 2.4.*

*To do this we determined most likely ice flow orientations for all ribbed moraine fields by calculating the angle transverse to landform orientation, and calculating the mean angle for each moraine field. We then compared these values to modelled ice flow orientation from Patton et al., (2016; 2017). This comparison was mode in timesteps every 1000 years, from 25 ka BP to 10 ka BP, resulting in 14 ka BP and 25 ka BP being two timesteps oriented most in line with mapped ribbed moraines.*

*This is not to say that all ribbed moraines mapped in this study are from the same timestep, as they likely formed at different times throughout deglaciation. Instead, we use the 21 ka BP timestep as within the data we have, this is the most commonly matching point in time for ribbed moraine field orientation, and ice flow orientation.*

**L235 - what is Shackletons theory? I am pretty sure this paper is an analysis of model output, not a theoretical proposal (i.e. one based in maths).**

*We have addressed this in response to reviewer 1's comments on this subsection, we no longer make reference to Shackleton for this point: "On re-reading, this makes sense, we have addressed this, another possibility is the Stokes et al., 2008 paper on ribbed moraines. Lines 273 – 276."*

*The "patchy" warm and cold based ice theory further connects with Kleman and Glasser, (2007) where subglacial thermal regime is strongly controlled by local variations in ice conditions, and Stokes et al., (2008) where they conclude ribbed moraines in ice streams to imply the presence of compressional "sticky spots" within ice stream flow.*

**Results - it is interesting that ribs seem to favour some places. But these seem to be basins - presumably just controlled by where sediment is available? For example, Figure 3, you aren't going to get ribs on the really steep slopes, as the sediment would fall off. This also impacts the hydraulic head calculation. As noted above, the model comparison is flawed.**

*We do not consider the relationship between ribbed moraines and areas that are "not steep" to be a causal relationship, simply a relationship. There is sediment available in places that are flat, but ribbed moraines do not form in all of them. With reference to slope for example, we see that ribbed moraines form in a band that is of lower slope than the mean slope of "non-ribbed moraine" pixels across Norway (Figure 3). Thus, there must be something more to it, for example a hydrological or ice dynamical factor. Furthermore, there are several sites across Norway where ribbed moraines do not form in basins, but flat, elevated regions – however a figure of this would be beyond the scope of this paper, as we are primarily focused on statistical analysis here, rather than spatial analysis. We have added a short point responding to this. Lines 318 – 322, S. 3.2.*

*This is notable as in all distributions, the maximum value for areas without ribbed moraines corresponds to the mean value for areas with ribbed moraines. Hence, we note ribbed moraines to be concentrated in areas of lower slope than Norway's average, but not outside the majority of values for Norway in general. This indicates a relationship between flat areas and ribbed moraine presence, but not necessarily a causal one, instead there are likely more factors than simply slope angle pertaining to the formation of ribbed moraines.*

**Section 3.1.1 - I agree there are problems with the resolution comparison. One could have offset this by calculating the difference between the model grid elevation, and the DEM elevation of the ribs, then applying this to the basal temperature calculation. The bigger flaw is taking one timeslice, as mentioned above.**

*We address the timeslice issue again here. Our resampling down to 250 m at least addressed the issue with hydrological and morphological data, and to a point will have given us some valley resolution in our outputs. We think that we have qualitatively described the issues with basal temperatures within the manuscript (Lines 466 – 478, S. 4.1.). We do agree that the point on calculating basal temperature would be possible, and would provide an estimate for basal temperature. However, this value would not be the true basal temperature, rather an estimation thereof, as it would disregard the effects discussed in the paper such as strain heating and also additional insulation due to thicker ice leading to reduced basal cooling. As such, as we cannot address all factors quantitatively, we maintain our qualitative approach here. The relevant lines (466 – 478, S. 4.1.) are included below.*

*We note that the glacier model we use for this study runs at a 10 km resolution, and thus detail smaller than 10 km is unresolved in our outputs. This is important, as ribbed moraine fields may often be longer than 10 km, but they typically are situated in depressions in the landscape (Dunlop & Clark, 2006), which are typically narrower than 10 km in width. These depressions, and by association, higher peaks than the average elevation of each grid cell, can lead to elevation variations by > 1000 m over the area of a single grid cell. As such, it is not unlikely that areas which are close in temperature to the pressure melting point (on average), may have a patchwork of deeper valleys where basal temperatures are at the pressure melting point, and thus basal water can be present. In addition to the spatial scale-derived inaccuracy, we note that the model used does not address frictional melting and strain-based heating processes within the ice mass, and thus the basal temperature in areas before obstructions in flow are likely lower than the true values. These two factors combine to present the possibility of there being water at the bed of the FIS, particularly in areas where ribbed moraines may have been common due to their presence in depressions at the glacier bed. Such effects from the variability within a 10 km cell may only just tip temperatures over the melting threshold, but this would be sufficient for a theory focusing on a patchwork of water-filled cavities.*

**L484 - What is glacial status?**

*Edited for clarity. Line 540, S. 4.2.*

**L580: There are no velocities in Ely et al., 2023.**

*We have addressed this discrepancy with Ely et al., (2023) throughout the paper.*

**Throughout the discussion there is some mention of a "cavity system" - what scale is this? Glaciologists would assume a 1-10 m scale as described by theories such as Kamb and Lliboutry. But it seems the authors are refering to km scale patches. Also, wouldn't the water be held in the sediment and thus mostly flowing via Darcian flow? I think this needs rethinking with sound glaciological basis.**

*We agree that there is likely water flow in the sediment, this provides a strong basis for several of the ideas discussed in Trommelen, Ross & Ismail (2014), however the hydrological system appears to have two potential formats. Either first, a system of many small cavities that are at the 10s of metres scale, but act as a larger body of water due to them being interconnected and wide-ranging, or secondly, something more akin to what we see in subaerial drainage networks, as suggested in Hall and van Boeckel, (2024; 2025). This could indicate some form of more major hydrological storage in the form of large, shallow subglacial lakes, forming a "larger" cavity. We have addressed this in-section now, with our primary suggesting being a large scale cavity field around saturated ribbed moraines. Lines 677 – 679 S. 4.4.2; Lines 684 – 688; S. 4.4.2.*

*Recent works suggest the potential for the subglacial drainage network being more similar to the subaerial drainage network, where subglacial lakes are not uncommon in topographic lows, thus providing the potential for water storage and a more interconnected subglacial hydrological network under high pressures (Hall & van Boeckel, 2024; 2025).*

*We also consider the opposite to be unlikely – that there were large amounts of subglacial flow, causing ribbed moraine to be formed in major outburst flooding (Fisher & Shaw, 1992), as the ice sheet model and calculated hydrological data do not describe sufficient subglacial water to compose a flood scenario. Yet, the temporal resolution, and recent studies into the subglacial hydrological system (Hall & van Boeckel, 2024; 2025) make it difficult to conclusively rule out a*

*flooding scenario, instead, we suggest the potential for a large-scale cavity network of interconnected cavities forming a large "cavity field" region around saturated ribbed moraines.*

**L611 - Both models show stable ribs, and don't account for cold-warm based transitions yet.**

*We have addressed this with a section added to this statement, specifically we refer to studies that suggest ribbed moraines to be transient landforms which evolve over time and under changing conditions, thus being related to patchwork thermal conditions. We also make reference to Ely et al., (2023), which avoids discussing changes in basal thermal conditions in the model used. Lines 670 – 676.*

*Whilst the Ely et al.'s (2023) model shows ribs being stable once formed, geomorphological evidence shows ribbed moraines transitioning into other bedforms such as drumlinoids (e.g. Möller & Dowling, 2018), as such it is likely that changes in ice conditions lead to evolutions in bedforms. Further, Putniņš & Henrikson (2017) determine ribbed moraines to be a product of changes in ice dynamic changes, whilst Stokes et al., (2008) advocate for "sticky spots", which can be viewed as the cold part of the cold-warm "patchwork". In combination, therefore, whilst models do not show ribbed moraines being transient, current models do not consider the basal thermal regime in detail (Ely et al., 2023).*

**Throughout the discussion of different processes is limited and needs thinking about. The instability theory for example does not prohibit shear-stacking from happening. It's just a process of sediment movement to this model.**

*The discussion of processes is limited in this paper, as this is beyond its scope, instead this paper aims to make observations of conditions and the relationships between these and ribbed moraines. We did not aim to assig process-based formational theories to ribbed moraines within this paper. Rather, we aimed to associate the occurrence of ribbed moraines with terrain and ice sheet conditions based on the current best modelling approach of the Fennoscandian ice sheet (Patton et al., 2016; 2017). These associations can be used to identify factors which could be used in future numerical modelling approaches, or geomorphological works to approach the formational processes of ribbed moraines.*

*Thanks for clarifying this point on shear-stacking. We only argue against the shear-stacking model in terms of its requirements for cold-based ice only, as this requirement doesn't appear to fit with the "patchwork" idea (Stokes et al., 2008), nor the numerical values indicating some degree of hydrological influence on ribbed moraine formation.*

**The conclusion - specifically conclusion F - seems to be that we can't rule out any of the hypotheses, therefore they are all likely correct. This reasoning is flawed. It should be that we can't rule any of the hypotheses, they could all be correct or all could be wrong, we need a better way of testing them.**

*We have edited the end of conclusion f – we agree that we require better data and methods for testing these hypotheses towards ribbed moraine formation. Our reasoning is made clearer, to state that all hypotheses are equally likely to be correct or incorrect, within the bounds of our findings in this study. Lines 746 – 751; S. 5; Lines 704 – 709, S. 4.4.2.*

**_Reference list:_**

Ahokangas, E., Ojala, A. E. K., Tuunainen, A., Valkama, M., Palmu, J. P., Kajuutti, K., and Mäkinen, J.: The distribution of glacial meltwater routes and associated murtoo fields in Finland, Geomorphology, 389, 107854, https://doi.org/10.1016/j.geomorph.2021.107854, 2021.

Aradóttir, N., Benediktsson, Í. Ö., Helgadóttir, E. G., Ingólfsson, Ó., Brynjólfsson, S., Farnsworth, W. R.: Ribbed moraines formed during deglaciation of the Icelandic Ice Sheet: implications for ice-stream dynamics, Boreas, https://doi.org/10.1111/bor.12690, 2024

Barchyn, T. E., Dowling, T. P. F., Stokes, C. R., and Hugenholtz, C. H.: Subglacial bed form morphology controlled by ice speed and sediment thickness, Geophys. Res. Lett., 43, 7572–7580, https://doi.org/10.1002/2016GL069558, 2016.

Barnes, T. J.: Inventory of Norwegian ribbed moraines, Zenodo, [dataset], https://doi.org/10.5281/zenodo.15496861, 2025.

Barnes, T. J., Filhol, S.: Aeteia/Ribbed-Moraine: Release ver.8.3 for ribbed moraines detection script, Zenodo [code and data set], https://doi.org/10.5281/zenodo.7991094, 2023

Barnes, T. J., Schuler, T. V, Filhol, S., and Lilleøren, K. S.: A machine learning approach to the geomorphometric detection of ribbed moraines in Norway, EGUsphere, 1–24, https://doi.org/10.5194/esurf-12-801-2024, 2024.

Ely, J. C., Stevens, D., Clark, C. D., and Butcher, F. E. G.: Numerical modelling of subglacial ribs, drumlins, herringbones, and mega-scale glacial lineations reveals their developmental trajectories and transitions, Earth Surf. Process. Landforms, 48, 956–978, https://doi.org/10.1002/esp.5529, 2023.

Finlayson, A. G. and Bradwell, T.: Morphological characteristics, formation and glaciological significance of Rogen moraine in northern Scotland, Geomorphology, 101, 607–617, https://doi.org/10.1016/j.geomorph.2008.02.013, 2008.

Finlayson, A., Merritt, J., Browne, M., Merritt, J., McMillan, A., and Whitbread, K.: Ice sheet advance, dynamics, and decay configurations: evidence from west central Scotland, Quat. Sci. Rev., 29, 969–988, https://doi.org/10.1016/j.quascirev.2009.12.016, 2010.

Fowler, A. C.: The philosopher in the kitchen, the role of mathematical modelling in explaining drumlin formation, GFF, 140:2, 93-105, https://doi.org/10.1080/11035897.2018.1444671, 2018.

Goudie, A.: Encyclopedia of Geomorphology, 2, Psychology Press, pp. 1013, ISBN 9780415272988, 2004.

Hall, A., van Boeckel, M.: Hydraulic damage in subglacial conduits: Evidence from rock hydrofracture and hydraulic jacking for high fluid pressures during rapid melt of the Fennoscandian Ice Sheet, Quaternary Science Reviews, 323, https://doi.org/10.1016/j.quascirev.2024.108917, 2024.

Hall, A., van Boeckel, M.: Debris production, transport, and sedimentation at high fluid pressures in subglacial conduits, Quaternary international, 720, https://doi.org/10.1016/j.quaint.2024.109650, 2025.

Hättestrand, C. and Kleman, J.: Ribbed moraine formation, Quat. Sci. Rev., 18, 43–61, https://doi.org/10.1016/S0277-3791(97)00094-2, 1999.

Kamleitner, S., Ivy-Ochs, S., Salcher, B., Reitner, J. M.: Reconstructing basal ice flow patterns of the Last Glacial Maximum Rhine glacier (northern Alpine foreland) based on streamlined subglacial landforms, Earth Surface Processes and Landforms, https://doi.org/10.1002/esp.5733, 2023.

Kleman, J., Glasser, N.F., The subglacial thermal organisation (STO) of ice sheets. Quaternary Science Reviews 26, 585–597. https://doi.org/10.1016/j.quascirev.2006.12.010, 2007.

Lindén, M., Möller, P., and Adrielsson, L.: Ribbed moraine formed by subglacial folding, thrust stacking and lee-side cavity infill, Boreas, 37, 102–131, https://doi.org/10.1111/j.1502-3885.2007.00002.x, 2008.

NGU: Løsmasser over Norge, https://www.ngu.no/om-geologi/om-losmasser, 2016.

Ojala, A. E. K., Mäkinen, J. K., Ahokangas, E., Kajuutti, K., Valkama, M., Tuunainen, A., Palmu, J-P.: Diversity of murtoos and murtoo-related landforms in the Finnish area of the Fennoscandian Ice Sheet, Boreas, 50(4), https://doi.org/10.1111/bor.12526, 2021.

Patton. H., Hubbard, A., Heyman, J., Alexandropoulou, N., Lasabuda, A. P. E., Stroeven, A. P., Hall, A. M., Winsborrow, M., Sugden, D. E., Kleman, J., Andreassen, K.: The extreme yet transient nature of glacial erosion, Nat Commun, 13, 7333, https://doi.org/10.1038/s4167-022-35072-0, 2022.

Patton, H., Hubbard, A., Andreassen, Auriac, A., Whitehouse, P. L., Stroeven, A. P., Shackleton, C., K., Winsborrow, M., Heyman, J. and Hall, A. M..: Deglaciation of the Eurasian ice sheet complex, Quat. Sci. Rev., 169, 148–172, https://doi.org/10.1016/j.quascirev.2017.05.019, 2017.

Patton, H., Hubbard, A., Andreassen, K., Winsborrow, M., and Stroeven, A. P.: The build-up, configuration, and dynamical sensitivity of the Eurasian ice-sheet complex to Late Weichselian climatic and oceanic forcing, Quat. Sci. Rev., 153, 97–121, https://doi.org/10.1016/j.quascirev.2016.10.009, 2016.

Ploeg, K., Stroeven, A. P.: History and dynamics of Fennoscandian Ice Sheet retreat, contemporary ice-dammed lake evolution, and faulting in the Torneträsk area, northwestern Sweden, The Cryosphere, https://doi.org/10.5194/tc-347-2025, 2025.

Schomacker, A.: Moraine, *Encyclopedia of Marine Geosciences*, Springer, pp. 1-6, 2014.

Sollid, J. L. and Torp, B.: Glasialgeologisk kart over Norge, 1:1000000, *Nasjonalatlas for Norge, Geografisk institutt, Universitetet i Oslo*, 1984.

Stokes, C. R., Lian, O. B., Tulaczyk, S., and Clark, C. D.: Superimposition of ribbed moraines on a palaeo-ice-stream bed: implications for ice stream dynamics and shutdown. Earth Surface Processes and Landforms, 33, 593–609, https://doi.org/10.1002/esp, 2008

Trommelen, M. S., Ross, M., and Ismail, A.: Ribbed moraines in northern Manitoba, Canada: Characteristics and preservation as part of a subglacial bed mosaic near the core regions of ice sheets, Quat. Sci. Rev., 87, 135–155, https://doi.org/10.1016/j.quascirev.2014.01.010, 2014.

Vérité, J., Ravier, É., Bourgeois, O., Bessin, P., Pochat, S.: New metrics reveal the evolutionary continuum behind the morphological diversity of subglacial bedforms, Geomorphology, 427, https://doi.org/10.1016/j.geomorph.2023.108627, 2023.